# Tracking microphysical variations in emissions from Karymsky volcano using MISR multi-angle imagery, and implications for volcanological interpretation

Verity J. B. Flower[1,2] and Ralph A. Kahn[1]

[1]Climate and Radiation Laboratory, Earth Science Division, NASA Goddard Space Flight Center, Greenbelt, MD 20771, USA

[2]Universities Space Research Association, 7178 Columbia Gateway Drive, Columbia, MD 21046, USA

*Correspondence to*: Verity J. B. Flower (verity.j.flower@nasa.gov)

**Abstract.** Space-based, operational instruments are in unique positions to monitor volcanic activity globally, especially in remote locations or where suborbital observing conditions are hazardous. The Multi-angle Imaging SpectroRadiometer (MISR) provides hyper-stereo imagery, from which the altitude and microphysical properties of suspended atmospheric aerosols can be derived. These capabilities are applied to plumes emitted at Karymsky volcano from 2000 to 2017. Observed plumes

from Karymsky were emitted predominantly to an altitude of 2-4 km, with occasional events exceeding 6 km. MISR plume observations were most common when volcanic surface manifestations, such as lava flows, were identified by satellite-based thermal anomaly detection. The analyzed plumes predominantly contained large (1.28 μm effective radius), strongly absorbing particles indicative of ash-rich eruptions. Differences between the retrievals for Karymsky volcano's ash-rich plumes and the

sulfur-rich plumes emitted during the 2014-2015 eruption of Bárðarbunga (Iceland) highlight the ability of MISR to distinguish particle types from such events. Observed plumes ranged from 30 to 220 km in length, and were imaged at a spatial resolution of 1.1 km. Retrieved particle properties display evidence of downwind particle fallout, particle aggregation and chemical evolution. In addition, changes in plume properties retrieved from the remote-sensing observations over time are interpreted in terms of

shifts in eruption dynamics within the volcano itself, corroborated to the extent possible with suborbital data. Particles emitted at Karymsky prior to 2010 contain the established sulfate proxy. After 2011, all

plumes contain similar particle components, but with varying light-absorption properties that correlate with timing relative to position within each respective eruption phase.

## 1 Introduction

Satellite-based remote sensing has become an integral part of ongoing volcano monitoring because financial and logistical constraints limit the widespread deployment of ground-based monitoring equipment at active volcanic sites (Sparks et al., 2012). Different remote sensing measurement techniques facilitate the monitoring of a variety of volcanic phenomena. Currently, infrared (IR), visible, and ultraviolet (UV) techniques identify and track volcanic plumes in near-real-time (Brenot et al., 2013). IR instruments, also used in near-real-time, identify surface thermal anomalies such as lava flows (Wright et al., 2002; 2004). However, remote sensing measurements have applications in fundamental volcanological research as well, and are particularly useful in assessing long term (> year) trends in volcanic activity (Flower & Carn, 2015; Murphy et al., 2013; van Manen et al., 2012), and variations resulting from geological setting, based upon the activity type (Flower & Kahn, 2017a).

In this work, we use the Multi-angle Imaging SpectroRadiometer (MISR) to analyze the emission altitude, dispersion characteristics and microphysical (0.001-10 μm) properties of suspended aerosols emitted by Karymsky volcano. MISR was launched aboard the NASA Earth Observing System (EOS) Terra satellite in December 1999, and is a visible-near infrared (VIS-NIR) passive sensor measuring upwelling shortwave radiation in four spectral bands (446, 558, 672 and 866 nm). The instrument consists of nine cameras positioned at nadir, plus four steeper angles (26.1°, 45.6°, 60°, 70.5°), in both the forward (F) and aft (A) viewing directions along the satellite orbit track (Diner et al., 1998). Through stereo matching of these images, plume altitude can be calculated from the observed parallax, with the added ability to correct for plume proper motion during the seven minutes it takes for all nine MISR cameras to observe a given location on Earth's surface (Moroney et al., 2002; Muller et al., 2002; Kahn et al., 2007; Scollo et al., 2010; Nelson et al., 2013). Unlike broad-swath instruments which offer near-global daily coverage, MISR has a measurement swath of ~380 km, limiting its use as a global near-real-time data source. However, MISR has additional capabilities that can provide important new

insights into atmospheric phenomena generated by volcanic activity. In particular, it uniquely provides stereo-derived plume heights and radiometrically retrieved aerosol-type constraints, and offers an operational data record over 17-years long. Previous work with MISR established plume observation rates in Kamchatka, characterized plume ejection altitude and linked the observations back to the geological settings of the Kamchatka region (Flower & Kahn, 2017a). Eleven volcanic plumes were identified by Flower & Kahn (2017a) originating from Karymsky and are incorporated into the current paper. Plume height and microphysical properties were also analyzed for the 2010 eruption of Eyjafjallajökull (Kahn & Limbacher, 2012) and for multiple years of activity at Etna (Scollo et al., 2010; 2012) using MISR data. Further, MISR profiling makes it possible to separate remobilized ash events from erupted plumes (Flower & Kahn, 2017b).

The remainder of this section details the location and eruptive history of Karymsky. Section 2 outlines data sources and the analysis techniques employed. The general characteristics of particles observed from Karymsky are discussed in Section 3. Inter-comparisons of plume properties over time, investigation of downwind plume development, and interpretation of observed plume differences in terms of activity within the volcano itself, are included in Section 4. Finally, conclusions are given in Section 5.

## 1.1 Karymsky volcano

Karymsky (Fig 1; 54.049°N, 159.443°E) is one of the most active volcanoes on the Kamchatka Peninsula (Russia) and has a summit elevation of 1.5 km above sea level (ASL) (van Manen et al., 2012). Located in the Eastern Volcanic Front (EVF) region (Fig. 1) of Kamchatka (Ponomareva et al., 2007), volcanic activity is driven by the subduction of the Pacific Plate (Manea & Manea, 2007). Karymsky has frequently been observed to produce small ash and gas explosions on time scales of minutes-hours (Johnson & Lees, 2000; Fischer et al., 2002). Eruptions have been defined as ranging from Strombolian to Vulcanian in nature (Lopez et al., 2013: 2015; van Manen et al., 2012), with variations attributed to magma recharge at depth (Fischer et al., 2002). Periodic larger eruptive events

have produced plumes extending up to altitudes of >6 km (Lopez, et al 2013; 2015; Flower & Kahn, 2017)

A review of traditional eruption reports, compiled by the Global Volcanism Program (GVP: http://volcano.si.edu), identified 348 eruptions during the operational lifetime of the MISR instrument to date. The review incorporates events where a definitive eruption date was reported, to prevent the inclusion of spurious reports. Therefore, the eruption catalog likely underestimates the exact number of plume emitted by the volcano. The review of traditional reports includes plume injection altitude over time. At Karymsky, 85% of all reported plumes were emitted at altitudes less than 5 km ASL. Almost half (45%) of plumes were observed between 3 km and 5 km ASL. The greatest number of reports in a single year occurred in 2011 (53 eruption reports), with two years containing 39 eruption reports each: 2008 and 2010 (Supplemental Data).

Recent field campaigns monitoring Karymsky with multi-parameter instrumentation (IR & UV imaging and infrasound) identified a variety of activity styles (Lopez et al., 2013; 2015). Unfortunately, these ground-based campaigns did not coincide with MISR observations and therefore direct validation is not possible. However, four dominant processes were observed at Karymsky: ash explosions, pulsatory degassing, gas jetting, and explosive eruptions (Lopez et al., 2013; 2015). Depending upon local observing conditions, MISR has the *potential* to observe the emissions from all of these processes. In practice, the observation of pulsatory degassing and gas jetting emissions by MISR is limited by their short residence time. Strombolian emissions generally have short atmospheric residence periods (< 1 day) as well (Ozerov et al., 2003), limiting observations from moderate temporal-resolution instruments such as MISR. The larger Vulcanian eruptions inject aerosol to higher altitudes (Newhall & Self, 1982) increasing plume entrainment into layers of atmospheric stability, which increases plume lifetime and dispersion (Kahn et al., 2007). The varying eruptive styles, and corresponding atmospheric residence time of emitted particles, significantly alias the ability of MISR to sample plume behavior. Vulcanian plumes are most conducive to MISR observing frequency, and make up the majority of MISR observations from Karymsky.

## 2 Methodology

### 2.1 Data collection

Fifteen plumes were identified by reviewing MISR data corresponding to eruption reports from the Global Volcanism Program database (http://dx.doi.org/10.5479/si.GVP.VOTW4-2013) and assessing all relevant MISR orbits using the MISR Browse Tool (http://l0dup05.larc.nasa.gov/MISR_BROWSE/orbit). The relevant MISR orbits were downloaded from the MISR data repository (https://l0dup05.larc.nasa.gov/MISR/cgi-bin/MISR/main.cgi), maintained by NASA's Langley Research Center (LaRC) Atmospheric Science Data Center (ASDC). One additional plume, observed on January 6, 2013, was characterized by low altitude dispersion indicative of a remobilized ash plume (Flower and Kahn, 2017b). Due to the restriction of this plume to inland regions, where aerosol-type determination is difficult, the retrievals are not included in this study.

Based upon the review of traditional eruption reports at Karymsky, the MISR observation rate is ~4%. The low MISR observation rate is a function of the regional overpass frequency (one observation every 2-3 days), the atmospheric lifetime of the emitted plumes, and cloud cover occurrence. At Karymsky, mild-moderate explosivity (e.g. *Strombolian*) leads to short atmospheric residence time (< 1 day) and consequently lower observation rates than for volcanoes characterized by consistently large explosive eruptions (Flower & Kahn, 2017a).

### 2.2 MINX Software Plume Classification

Eruption plumes from Karymsky (2000-2017) were processed with the MISR INteractive eXplorer (MINX) program (Nelson et al., 2008; 2013). MINX is a stand-alone software package developed at the NASA Jet Propulsion Laboratory (JPL) and distributed through the Open Channel Foundation (http://www.openchannelsoftware.com/projects/MINX). The program derives plume elevations from MISR hyper-stereo data for a range of natural and anthropogenic processes, including volcanic eruptions. Stereo matching in both the red (672 nm) and blue (446 nm) spectral-bands facilitates analysis of drifting plumes over both water (where the higher-pixel-resolution red channel is generally preferred) and land (where the blue imagery usually shows plume features more distinctly). The MISR red-band data have 275 km spatial resolution in all the off-nadir cameras, whereas the blue band offers

1.1 km pixel resolution at all MISR camera angles (except 250 m at nadir). Where red-band contrast in plume imagery is sufficient, we take advantage of the improved horizontal and vertical resolution. Corrections are made within MINX to account for plume proper motion between the observation times of each camera (<7 min), from which wind speeds influencing plume dispersion are calculated (Supplemental Data).

The MINX output provides height retrievals across the plume length, allowing interpretation of plume dispersion dynamics and external influences affecting plume dispersion (Fig. 2; 3; 4). Plume heights from MINX are also used in the comprehensive particle-type analysis conducted with the MISR Research Aerosol retrieval algorithm (RA; see section 2.3).

## 2.3 MISR Research Aerosol Retrieval Algorithm

The MISR RA (Kahn et al., 2001; Limbacher and Kahn, 2017) was used to derive particle types for the fifteen volcanic plumes generated by Karymsky volcano during the study period (Table 1). Prior to particle property interpretation, MISR camera angles were co-registered to the plume height defined by MINX plume height retrievals. Where the plume displayed a significant change in altitude across its length, the plume was co-registered over a series of heights appropriate to each segment, and the individual segments were recombined in the interpretation stage of the analysis.

The MISR RA incorporates radiance data from each of the nine MISR cameras. These data represent the backscatter properties from a range of view angles. As particles scatter light differently across viewing angles depending upon their relative size, shape and light absorption, the properties of particles in a plume can be discerned if their MISR radiance signature is unique, and known. After incorporating the relevant MISR data and co-registering the plume imagery, the MISR RA compares the observed radiances with those simulated for a 774-mixture look-up table (Limbacher & Kahn, 2014; 2017), to match the microphysical properties of imaged aerosols with candidate mixtures. The 774 mixtures are comprised of three aerosol components, combined in varying proportions from among 14 component options. For this study, RA retrievals were performed at 1.1 km horizontal resolution over the plume area. The MISR RA selects aerosol optical models (aerosol types) and corresponding aerosol column amount (Aerosol Optical Depth; AOD) that produce acceptable matches to the MISR-observed multi-

camera top-of-atmosphere equivalent reflectances. Confidence in the aerosol-type retrievals increases when the mid-visible AOD (558 μm) exceeds about 0.15-0.2 and when the components of interest contribute at least ~20% of total AOD (Kahn & Limbacher, 2012; Kahn et al., 2001, 2010; Kahn and Gaitley, 2015). Based on the RA retrievals we define particles within three loosely defined size classes,

distinguished by their effective radius ($r_e$): Small ($r_e$ - 0.06 – 0.3 μm), Medium ($r_e$ – 0.3 – 1.0 μm), Large ($r_e$ – 1.0 – 10 μm). The maximum retrieved particle size is limited by the wavelength range of MISR observations (446 – 866 nm).

In the current RA aerosol-type climatology, the available spherical, absorbing component particles have a uniform effective radius of 0.12 μm, and mid-visible Single Scattering Albedo (SSA) of 0.8 or 0.9.

Particle absorption is represented by either flat or steep absorption spectral dependence. 'Flat' spectral absorption features absorb uniformly between spectral channels and represent particles that behave more like pollution particles. 'Steep' profiles are characterized by greater absorption at shorter wavelengths and tend to be better optical analogs for wildfire smoke (Chen et al., 2008; Limbacher and Kahn, 2014).

The current algorithm climatology lacks good optical models for non-spherical volcanic ash particles, and is restricted by the maximum MISR wavelength of 866 nm; therefore the retrievals have limited ability to distinguish variations in larger plume particles (1.5 – 10 μm). The algorithm tends to pick mixtures of component optical analogs from the climatology that ***in combination*** represent the scattering properties of the plume particles reasonably well (Kahn and Limbacher, 2012). The identified

variations in the available modeled mixtures can subsequently be used to qualitatively assess changes in particle properties across individual plumes. Comparisons between plumes were also performed to track whether particle properties varied during the analysis period (2000-2017).

To interpret plume characteristics, retrieved particle analogs were grouped into ~10 km$^2$ grids (9x9 1.1 km MISR RA retrieval regions). The proportion of aerosol types (size, shape and absorption) within

each grid was assessed to identify particle development both in individual plumes and between eruptions. The grouping of individual retrievals in this manner reduces the relative influence of any individual spurious mixtures that might be produced by the retrieval algorithm.

In order to establish whether volcanic plumes were distinguishable from meteorological clouds, retrievals were also collected from cloud features in the analyzed images. Retrievals during the analysis period (2000-2017) produced fairly uniform results, comprised of: ~90% large, spherical, non-absorbing particle optical analogs, and ~5% small spherical absorbing (flat profile) components, with the remaining fraction identified with the "medium dust grain" non-spherical component (Table 2).

Interference factors exist, limiting the techniques employed. The most common factor preventing plume retrieval is meteorological cloud cover. Clouds most often prevent the retrieval of low-altitude volcanic plumes underlying meteorological clouds, because the MISR VIS-NIR measurements cannot penetrate dense cloud cover. Volcanic features that exceed the altitude of local meteorological cloud tops can be analyzed for plume height with MINX. However, significant cloud cover also affects the application of the MISR RA. High surface reflectance can also impede RA analysis, especially in winter months, when Kamchatka has nearly complete snow cover. In winter, when both cloud cover and surface reflectance are not conducive to MISR analysis, retrievals were obtained only over water. Near-coastal regions were also analyzed with caution during winter, as the development of sea ice can impede plume analysis. In this work, sea ice impacted analysis of three plumes (P2011a, P2011b, P2011c; see Table 1 for plume designations), with one prevented entirely (P2011b).

## 3 Overview of MISR Retrieved Plumes Properties

### 3.1 Plume observation and dispersion characteristics

Karymsky has been active throughout MISR's operational lifetime (2000-present), with plumes observed in each of ten years (Supplemental Data). The greatest number of plumes identified by MISR in a single year occurred in 2011 with five events, three of which were observed in a nineteen-day period (31 Jan – 18 Feb). The increase in MISR-observed plumes in 2011 corresponds to an increase in the number of traditional plume reports in this year (Section 1.1; Supplemental Data). Imaged plumes varied in horizontal extent (30 – 220 km) and dispersion characteristics (Fig. 2; 3; 4). Plumes were broadly classifiable into four dispersion categories: large plumes, generally stable in altitude with

distance from source (P2007b, P2011a, P2013); large dense plumes displaying uplift or bifurcation with distance from source (P2007a, P2011d, P2011e, P2015); complex, multi-pulse plumes (P2004, P2006, P2011b); and smaller, diffuse plumes (P2003, P2005, P2011c, P2016). The remaining plume (P2009) displays low altitude dispersion similar to remobilized ash (Flower & Kahn, 2017b), discussed further

in Sect. 4.1. There is no apparent temporal pattern to the characteristics of plume dispersion and the overall timing of eruptive events. Plumes were most frequently emitted to an altitude of 2-4 km ASL, with occasional events exceeding 6 km. These average MISR-determined plume altitudes also correspond to the average plume height determined through the traditional report review (see Section 1.1; Supplemental Data). Ambient meteorological conditions were observed to influence downwind

dispersion, including the uplift of the plumes from initial injection altitude due to local frontal system dynamics (e.g. Fig. 7b). Flower & Kahn (2017a) previously established that plumes emitted from volcanoes in southern Kamchatka were also strongly influenced by the local atmospheric stability profiles.

### 3.2 MISR Micro-Physical Particle Properties

The MISR RA was run on the Karymsky plumes (excluding P2011b), and of the 774 mixtures available, combinations of only 100 mixtures were retrieved (see Supplemental Data). Plumes were comprised of just nine of the 14 components in the algorithm climatology (Table 2), indicating considerable particle-type selectivity. The mixtures selected exhibit a dominance (73%) of medium, non-spherical, weakly absorbing (MeNspWab dust-grain optical analog) components. Over half (58%)

of selected mixtures also contain the large, spherical, non-absorbing (LaSpNab) particles. Only six of the retrieved mixtures contain neither of these two components. These six are dominated by small particles with varying levels of absorption. Taken together, plumes at Karymsky are predominantly characterized by large (indicated by low Angstrom Exponent; Fig 5c), highly non-spherical (Fig. 5e), strongly absorbing (Fig. 5g) particles.

Due to the relative uniformity of the plume aerosol types retrieved at Karymsky, a secondary test case was analyzed, for eruptions dominated by sulfates. The Bárðarbunga (Iceland) eruption of 2014-2015 produced significant quantities of sulfur dioxide (Schmidt, et al., 2015), and due to the high latitude of

the volcano, three plumes were imaged by MISR (Supplemental Data). The MISR RA analysis of the Icelandic plumes (e.g. Fig. 5) display characteristics distinct from those produced at Karymsky. The sulfur-rich plumes of Bárðarbunga contain small (indicated by high Angstrom Exponent; Fig 5d), highly spherical particles (Fig. 5f) with minimal absorption (Fig. 5h). The contrast between plume

properties from Karymsky and Bárðarbunga, supports the conclusion that Karymsky plumes are predominantly ash-rich.  This conclusion also supports previous MISR analysis of multiple eruptions at Mt. Etna, where ash-dominated and sulfate-dominated plumes were successfully distinguished (Scollo et al., 2012).

**4 Detailed investigation of MISR Retrieved Plumes Properties**

The majority of Karymsky plumes were dominated by of the 'large' particle class (LaSpNab). Differentiation of large particles in the MISR RA is limited by the lack of good optical models for non-spherical volcanic ash particles, and the lack of MISR wavelengths exceeding 866 nm. Despite the limitations, and the relative consistency of Karymsky plumes, particle property differences were

characterized qualitatively for the imaged plumes. Changes were analyzed based on the relative variations in particle size, the fraction of smaller, angular particles, and the levels of light-absorption retrieved. Although general differences are observed between particle components (e.g. particles size, light-absorbing fraction), systematic co-variation between the two dominant aerosol components is apparent when plume retrievals are considered individually (Fig. 6g), suggesting that the variations

observed in the relative fractions of the remaining components are generated by changing plume properties and not by the particle selection process imposed by the algorithm. The fraction of large, spherical, non-absorbing (LaSpNab) particles is consistently anti-correlated with changes in medium, non-spherical, weakly absorbing (MeNspWab) particles (Fig. 6d). The inverse correlation between MeNspWab and LaSpNab particles was strongest ($R^2 = 0.95$) when plumes contained minimal amounts

of medium, spherical, non-absorbing (MeSpNab) particles (Fig. 6g).

At Karymsky, three distinct regimes are apparent when the plume properties are stratified by the AOD proportion of medium, non-spherical, weakly absorbing particles (MeNspWab) in each retrieval (Fig.

6a): Class 1 - low MeNspWab (<0.04); Class 2 - moderate MeNspWab (0.05-0.15); and Class 3 - high MeNspWab (0.19-0.31). Smaller particles displayed an increase between Class 1 and 2 but show stabilization between Class 2 and 3 (Fig. 6b). Aggregated absorbing component fraction was observed to increase across class (Fig 6e). The designated classes also display distinct levels of sphericity, driven by the co-variation of the large spherical and medium non-spherical components (Fig 6f).

The proportional changes in small, medium and large particles (Fig. 6b; c; d) means that, in aggregate, as the plume Class increases, plume particles decrease in size. Also, as Class increases, the plumes become more absorbing (Fig. 6e) and less spherical (Fig. 6f). The relative fraction of absorbing particles was observed to increase as Class increases. However, the absolute proportion of the small, highly and moderately absorbing particles (SmSpHab(f), SmSpHab(s) and SmSpMab(f)) remained consistent (<0.1 of total $AOD_{558}$) irrespective of the class. Therefore, the increase in absorption appears to be driven by the addition of weakly absorbing grains (Fig 6e). Plume 2007b represents a significant outlier in the analysis of Karymsky (single dot in Fig. 6b-d): P2007b was dominated by medium particles (Fig. 6c), with minimal small-particles (~3%; Fig. 6b) and no large components (Fig. 6d).

Plumes at Karymsky display similar fractions of spherical particles (0.65-1) to those observed in previous MISR analysis at Etna (Scollo et al., 2012). However, although retrievals at Etna were dominated by small, spherical particles analogs, Karymsky tended towards larger spherical particles. The larger particle analogs retrieved at Karymsky are likely proxies for fine ash, which is not captured explicitly among the MISR RA optical models. Where plumes are dominated by larger ash particles, the lack of large, non-spherical, absorbing components within the algorithm climatology restricts mixture selection. Plumes with these characteristics would be retrieved as combinations of the largest spherical particles (LaSpNab), with a fraction of smaller, absorbing components (SmSpHab(f), SmSpHab(s) and SmSpMab(f)) and medium grains (MeNspWab) to offset the spherical, non-absorbing qualities of LaSpNab (Kahn and Limbacher, 2012). Plumes imaged after 2010 appear to be dominated by these ash-proxy characteristics. The variations in small-medium analog fractions retrieved at Karymsky and Etna are likely the result of actual differences in eruptive characteristics. Etna is a stronger and more consistent emitter of $SO_2$ (~2000 tons/day) than Karymsky (~900 tons/day) (Carn et al., 2017). The

variations in particle size must be considered within the fraction of spherical particles for this to be used as a guide for the qualitative assessment of plume composition. *None* of the Karymsky plumes would be considered 'sulfur/water dominated' as defined by Scollo et al. (2012). However, if we use the comparative analysis of Bárðarbunga eruption of September 25, 2014 (Fig. 2b) to determine a sulfate proxy (MeSpNab), five plumes indicate the presence of small-moderate levels of sulfates (P2003, P2004, P2006, P2007a, P2009). Plume P2007b was identified as being dominated by this mixture (Fig. 7) combined with medium grains (MeNspWab). These observations represent a partial validation of the capabilities of MISR volcanic plume retrievals as qualitative indicators of changes in particle type, opening the possibility of interpreting the variations in plume properties over time in terms of the volcano's geological activity. However, to validate the ability of MISR to track these changes in plume properties, coincident *in situ* or ground-based observations are required.

## 4.1 Plume observation and dispersion characteristics

The 1.1 km spatial resolution of the MISR RA retrievals makes it possible to investigate the evolution of plume particle properties within individual plumes as dispersion occurs downwind. Figure 7 shows along-plume aerosol-type retrievals for an example of each of the three retrieval plume-particle-property classes (Fig. 6) identified at Karymsky. For comparison, an additional retrieval is included for the sulfate-rich case from Bárðarbunga volcano that is highlighted in Figure 5. For the Class 1 plume (P2011a), the retrieved aerosol components remain relatively constant along the length of the plume (Fig. 7a). Class 2 plumes contain comparatively smaller particles (higher fraction of MeNspWab) and show increasing absorption with distance from the source (MeNspWab and SmSpHab(f); Fig 7b). The Class 3 plume (Fig. 7c) is predominantly comprised of the medium, spherical, non-absorbing (MeSpNab) component, likely to represent a sulfate proxy. The medium, non-spherical 'dust' analog (MeNspWab) component is also present, suggesting that despite having more sulfate, the plume contained a significant fraction of ash particles. The sulfate-rich retrieval (Fig. 7d) at Bárðarbunga displays an overwhelming dominance of the MeSpNab component, with minimal fractions of small, spherical particles (SmSpNab). This plume also shows diminishing absorption downwind from the near-source region, possibly indicating chemical or physical conversion within the plume. Alternatively,

emitted particle properties from the volcano could be varying over time, as the MISR plume observations represent snapshots of material emitted over a longer period. However, the consistency of downwind particle property changes tends to favor the particle-evolution interpretation, as the alternative would require the snapshots of multiple plumes to have coincidentally captured the same sequence of erupted particle differences.

Influences likely to affect the development of plume properties downwind include particle fall-out, chemical conversion, particle hydration, and physical aggregation of particles within the atmosphere. In the case of particle fall-out, we would expect the fraction of larger particles to diminish with distance from the source (e.g. Fig 7b), as observed in five Karymsky plumes (P2005, P2007a, P2011c, P2011d, P2013; Supplemental Data). In contrast, particle aggregation would lead to a reduction in smaller particle fraction with distance, corresponding to an increase in medium and larger particles, observed in three other plumes (P2003, P2004 and P2016; Supplemental Data). In multiple retrievals (see Supplementary data) we see changes in the levels of absorption with distance from the source. In several of these cases (P2003, P2009, P2011c and P2013; Supplemental Data), the near-source region is characterized by a higher fraction of spherical absorbing (SmSpHab & SmSpMab) particles (e.g. Fig 7d). Variations in the plume particle light-absorption properties likely provide the best remote-sensing evidence for chemical processing or particle hydration. In all cases where spherical absorbing particles (SmSpHab and SmSpMab) were observed, the more absorbing (SmSpHab) mixture tends to dominate nearer source (<40 km) regions (Supplemental Data). A similar result was derived from MISR for the 2010 Eyjafjallajokull eruption, and was interpreted as likely representing the alteration of absorbing particles by oxidation and/or hydration as the plume interacts with the ambient atmosphere (Kahn and Limbacher, 2012).

To verify deductions about the processes influencing plume development inferred from remote sensing, detailed compositional information from suborbital plume observations would need to be made, preferably near the MISR overpasses time. Existing ground-based observations by Lopez et al. (2015) did identify some shifts in the emitted particle properties over time. In surface-based observations of an

eruption occurring over ~1-minute time frame, the plume was initially comprised of larger particles (~13 μm), transitioning to smaller components (~2.5 μm), and finally to particles of increased size again (>20 μm), in this case mostly coarse-mode particles, near or beyond the sensitivity limit of the particle sizing analysis technique employed here (Lopez et al., 2015). MISR would tend to retrieve the larger

particles as LaSpNab (0.013-8.884 μm), whereas the smaller particles would likely be retrieved as medium, weakly absorbing grains (0.1-1.0 μm), depending upon the specific characteristics of the plume. A MISR observation of this event would likely contain a fraction of LaSpNab and MeNspWab. Ongoing variations in the erupted products, such as those observed by Lopez et al. (2015), would subsequently lead to varying mixtures selected by the MISR RA along the plume length. The injection

altitude, and corresponding wind speed, would help determine the segregation of these varying eruptive products as dispersion occurs. The specific nature of particle-property differences, induced by eruption history and/or altered by ambient conditions, significantly impacts the ability to detect particle-type changes from 1.1 km MISR retrievals. Our dominant particle retrievals (Fig. 6) appear to correlate well with the *expected* within-plume structure obtained from ground-based observations at Karymsky (e.g.,

Lopez et al., 2015). Yet, the observed transitions could at least in part be the result of variations in volcano emissions over time, rather than solely due to particle evolution downwind. To definitively separate variations in emitted particle properties from downwind conversion, ground-based observations coinciding with MISR observations are required.

**4.2 Variations in plume properties over time, and volcanological implications**

MISR-derived plume particle types and dispersion dynamics for eruptive events at Karymsky also varied over time (Fig 8). These variations are indicative of cyclical processes on the timescale of months to years within a continuously evolving magmatic system during the analysis period (2000-2017). This observation corroborates existing ground-based and alternative satellite-based observations

of Karymsky activity (van Manen et al., 2012; Lopez et al., 2013). The majority of MISR-observed plumes occur during periods when pixel-integrated radiant temperature, as measured by MODIS (Fig. 8), is above background levels. These MODIS thermal anomalies reflect the presence of lava flows,

lava lakes, and to a lesser extent, lava domes and pyroclastic flows (Wright et al., 2002; 2004) and provide ~4 observations/day of Kamchatka from two operational satellites (Terra & Aqua). The presence of thermally radiating surface manifestations of volcanic activity can be used to infer changes in system dynamics. The eruption characteristics at Karymsky inferred from thermally radiating features

suggests the separation of the eruption into phases interspersed with periods of dormancy (Fig. 8). The defined eruption phases were chronologically assigned numerical values to distinguish each event. Where previous work by van Manen et al., (2012) denoted eruption phases, additional notations have been made (e.g. SvM1) in Figure 8.

*Pre-2008*

Van Manen et al. (2012) determined that phases in the thermal anomaly record formed three distinct patterns of eruption development at Karymsky. Six plumes, imaged by MISR, correspond to the pre-2008 time period investigated by van Manen et al. (2012). These plumes (P2003, P2004, P2005, P2006, P2007a and P2007b) display variations in particle properties that appear to correlate with shifts in the

dynamics of thermal anomaly development identified by van Manen et al., (2012). Plumes P2003 and P2004 occurred in an eruption phase characterized by consistent thermal anomalies over time. Both of these plumes fall into Class 1 (low – MeNspWab; Table 1), suggesting that plumes generated under this regime display similar microphysical particle properties. The P2005 plume was observed just before the beginning of a new thermal anomaly phase, and this plume displays lower- MeSpNab, (Class 3),

indicative of a less sulfate-rich plume. GVP documents an ash-rich plume emission on August 27 2005, less than 12 hours prior to the MISR observation, suggesting an ongoing ash-rich eruption.

Plume P2006 falls into a waning eruption phase, as thermal output steadily decreased from Karymsky during the four months leading up to the observed plume. This plume is characterized by low MeNspWab (Class 1), but contains a lower proportion of sulfate/water proxy (MeSpNab) than the 2003-

2004 events. Lower sulfate/water content combined with the corresponding decrease in thermal anomaly data (Fig. 8; van Manen et al., 2012) suggests that the plume resulted from decreased magma volatile content and subsequently higher magma viscosity. In these situations, increasing magma viscosity leads to the formation of a dense plug sealing the volcanic conduit (Clarke et al., 2015). These

capped volcanic vents are then subjected to extreme pressure build up below the plug, until its mechanical strength is exceeded (Self et al., 1978). The over pressurized system forcefully ejects trapped gas and fragmented magma leading to ash rich plumes. The continued reduction in thermal anomaly detection following P2006 (Fig. 8) would suggest that less volatile-rich magma was entering the system from depth, leading to a waning of the eruption phase. The thermal anomaly record indicates a final increase (Fig. 8) in thermal output, prior to the termination Eruption Phase 2 in December 2006.

The final two pre-2008 plumes (P2007a and P2007b) correspond to a single eruption phase in the thermal anomaly record in 2007, characterized by a rapid increase in the pixel-integrated temperature up to the saturation level (van Manen et al., 2012), after which the eruption ceased abruptly (Fig. 8). One plume (P2007a) was observed within a month of the initiation of this phase, and the second (P2007b) within a month of its termination. The earlier plume contained a moderate proportion of medium, non-absorbing grains (MeNspWab) and minimal levels of the sulfate/water proxy (MeSpNab). In contrast, P2007b shows a dominance of medium, spherical, non-absorbing particles (MeSpNab) and a moderate fraction of medium, weakly absorbing grains (MeNspWab). The increase in the proportion of spherical, non-absorbing particles between P2007a and P2007b, combined with the constant extrusion of material at the surface (thermal anomalies in Fig. 8), suggests that the system, was initially cleared of viscous, volatile-poor magma (P2007a), causing depressurization and upwelling of volatile-rich magma driving the ongoing eruption. The constant emissions in 2007 likely also led to the depletion of the feeding magma, accounting for the subsequent, abrupt cessation of this eruption phase.

*2008-2010*

The single plume identified during this period occurred on April 26, 2009 (P2009; Fig. 8) displays bi-modal plume altitude (Fig. 3d). The majority of the plume exhibits the low-altitude, limited buoyancy characteristics of remobilized ash events (Flower & Kahn, 2017b), and a secondary lofted plume is found at ~2 km. Remobilized plumes in the Kamchatka region were consistently dominated by very large, non-spherical particles, which was not the case in this instance. Higher levels of light absorption suggest that this Karymsky plume was the result of a low-altitude, limited buoyancy eruption, as distinct from a remobilized ash event. According to the traditional eruption reports, despite occurring during a

period of relative quiescence, a weak thermal anomaly was observed over Karymsky on April 26, 2009 (KVERT, 2009). The combination of a low altitude, topographically defined plume and the occurrence of a short-lived weak thermal anomaly suggest that this plume might have been produced by a small pyroclastic flow. A pyroclastic flow resulting from a small depressurization of the vent could have

caused the emission and suspension of volcanic material in the lower atmosphere (Malin & Sheridan, 1982; Sulpizio et al., 2014). An event of this nature would lead to the exposure of underlying, warmer magmatic material at the surface, producing a weak thermal anomaly (Wright et al. 2002; 2004). Unfortunately, there are no ground-based reports from Karymsky during this time to validate this interpretation.

More generally, comparison of whole plume component fractions (Fig. 8; Supplemental Material), beginning in 2003, highlights a shift in retrieved particle properties toward the end of this period. Prior to 2011, plumes exhibit small-moderate (<55%) fractions of the MeSpNab component. Investigation of known sulfate/water-rich plumes, such as that emitted from Bárðarbunga volcano (Schmidt et al., 2015) and analyzed with the MISR RA, were consistently retrieved as 55-95% MeSpNab, with minor

contributions (<10%) of SmSpNab (Fig. 8 Inset). This supports our use of MISR RA-retrieved MeSpNab as a qualitative proxy for sulfate-rich plumes. Based on this proxy, we infer that earlier plumes (2003-2010) contained higher fractions of sulfate compounds than those emitted in subsequent years (2011-2017).

***2011-2017***

In addition to a decrease in sulfate-analog AOD fraction, post-2010 plumes also show increasingly steep absorption, not present earlier in the analysis period. A shift in the absorbing characteristics of particles from exclusively flat spectral dependence to relatively equal fractions of flat and steep spectrally varying particle absorption could reflect a change in the composition of erupted products. Plumes

emitted from 2011 also consistently display plume components (LaSpNab, SmSpHab(f), SmSpHab(s), SmSpMab(f) and MeNspWab), but in varying proportions, suggesting an evolution of emitted particles over time. Plumes observed in early 2011 (P2011a and P2011c) are composed of >90% large, spherical, non-absorbing components (LaSpNab), with small proportions of small and medium absorbing

particles. By mid-2011 (P2011d and P2011e), the large component (LaSpNab) drops to ~78%, with corresponding increases in the medium non-spherical and the smaller, spherical, absorbing constituents (MeNspWab, and SmSpHab(f), SmSpHab(s), SmSpMab(f)). The transition from large components to medium, non-spherical grains (MeNspWab) suggests a shift to finer ash particles as the eruption progressed. All subsequent plumes in the satellite record through 2017 (P2013, P2015 and P2016) display the same particle components to those observed later in 2011.

The timing of eruptions, relative to the phase start date (Fig. 8), appears to control the relative fractions of the absorbing particles observed within each plume. The normalized 'eruption day' for each MISR plume is calculated in days after the eruption start-date. The eruption start-date is defined as the earliest thermal anomaly detection by MODIS in a particular phase. These normalized timings were used to track changes in the AOD fraction of absorbing particles (SmSpHab(f), SmSpHab(s), SmSpMab(f) and MeNspWab) throughout the 2011 phase (Fig. 6h); this quantity displays a strong second-order polynomial relationship with eruption day ($R^2 = 0.999$). When P2013 and P2015 are also normalized by eruption day, the correlation remains high ($R^2 = 0.995$). The strong eruption day/component fraction correlation suggests that processes occurring between 2011 and 2015 represent an equilibrium state within the volcano. As such, eruption characteristics develop consistently over time within each distinct phase. The stabilization of plume properties is likely linked to the waning of the eruption, evidenced by decreasing thermal output (Fig. 8), leading to eruption termination in 2016. *In situ* measurements, if available, would help validate, and add detail to, the physical developments occurring at the volcano during this time.

The interpretation of eruption dynamics, based on the inter-comparison of MISR-derived microphysical plume properties and the MODIS surface thermal radiance record, highlights the value of using complimentary, remote-sensing datasets to assess eruptive dynamics. Extension of the current technique to include additional, complimentary remote sensing data such as $SO_2$ concentrations from the Ozone Monitoring Instrument (OMI; e.g., Carn et al., 2017) would facilitate a deeper understanding of volcanological processes.

**5 Conclusions**

This paper demonstrates the depth and versatility of volcanic plume studies made possible with the combination of MISR multi-angle and MODIS thermal anomaly data. We analyzed Karymsky volcano plume dispersion dynamics based on near-source vertical profiling from MISR hyper-stereo imagery.

Additionally, we constrained volcanic plume particle size, sphericity, and light-absorbing properties from the MISR observations, and, with the help of MODIS remotely sensed thermal anomalies, traced aspects of volcanic eruption evolution over the 17-year data record. MISR retrieval climatology is limited by a lack of good optical models for non-spherical volcanic ash particles and the maximum observed wavelength (866 nm) limits sensitivity to particles larger than a few microns in diameter.

Despite these limitations, we derived constraints on aerosol properties that reflect qualitative changes in aerosol type both within individual plumes and between eruptions over time. These observations, in turn, can be interpreted in terms of eruptive style (sustained emission, pulsatory explosivity, plume buoyancy) and downwind plume particle evolution.

Plume particles from the 15 observed Karymsky eruptions were dominated by the largest available component in the retrieval algorithm climatology (LaSpNab; 1-10 µm size range; $r_e$ - 1.28 µm). Overall, plumes fell into three broad categories, distinguished by the AOD fractions of medium, non-spherical, weakly absorbing grains (MeNspWab; $r_e$ - 0.75 µm), and smaller, spherical, absorbing particles ($r_e$ - 0.12 µm). An inverse relationship was observed between large, spherical, non-absorbing

and medium, weakly absorbing 'grain' AOD fractions, representing shifts in plume properties between relatively fine ash (1.28 µm effective radius) to very fine ash (0.75 µm effective radius). Further, comparison of retrievals from Karymsky with known sulfate-rich eruptions at Bárðarbunga, Iceland demonstrate the ability of MISR to discriminate between sulfate-rich eruptions from more ash-rich events. In some cases, changes in retrieved particle properties were observed downwind, indicating

possible particle coagulation, large-particle fall-out, and particle brightening, likely due to chemical weathering or hydration.

Although the majority of MISR observed Karymsky plumes were determined to be predominantly ash-rich, some variation was observed among events. In more detail, particle properties varied throughout the observation period (2000-2017) in a manner consistent with moderate (months-years) cycles within the ongoing decadal eruption evolution. From 2003 to 2007, plumes were dominated by the larger particles, with a moderate fraction of the sulfate analog. Particle property variations appear to correlate to changes in the eruption dynamics of individual phases. In 2007, a single eruption phase indicates a shift toward more sulfur-rich, smaller-sized particles as thermal output increased, from which we infer the ascension of more volatile-rich magma, culminating in multiple eruptions, magma depletion, and finally, rapid termination of the eruption cycle. MISR captured only one low-elevation plume at Karymsky between 2007 and 2011, likely resulting from a small dome collapse and pyroclastic flow event.  Plumes generated in 2011 indicate a stabilization of volcanic processes leading to predictable particle evolution throughout the eruption phase. These developments suggest a reduction in the volatile content and alteration of the chemical composition of the feeding magma. The level of absorbing components appears to correlate strongly with normalized eruption day, assessed relative to the earliest thermal anomaly observed during a particular phase; this metric facilitates comparison between eruption phases. Plumes emitted in 2013 and 2015 also appear to follow this pattern of particle development relative to the normalized eruption start date. The consistency of eruption development suggests that the magmatic feeding system at Karymsky stabilized as the eruption waned (2011-2016).

Deductions based upon space-based remote-sensing observations, corroborated by suborbital measurements where possible, illustrate the ability of such data to be applied for broader volcanology-from-space applications. As the satellites provide frequent coverage, there are considerable opportunities for further study, even of remote locations around the globe where *in situ* monitoring is lacking.

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

**Table 1: MISR observed volcanic plumes from Karymsky during the study period (2000-2017).**

| Date | Orbit [*Plume #*] | Path | Block | MISR Observation Notes* [Particle Type Category[†]] | |
|------|-------------------|------|-------|-----------------------------------------------------|--|
| 12/27/2003[a] | 21402 [*P2003*] | 99 | 47 | Moderate plume (30 km) dispersing E at an altitude ~2 km [1] | |
| 01/12/2004 | 21635 [*P2004*] | 99 | 46-47 | Large (80 km) multi-segment plume dispersing ENE ranging in altitude ~2-4 km [1] | 5 |
| 08/28/2005 | 30285 [*P2005*] | 97 | 47 | Moderate plume (30 km) dispersing ESE at ~3 km [3] | |
| 10/23/2006 | 36416 [*P2006*] | 100 | 47-48 | Large plume (>80 km) with complex dispersion E–NE–N increasing in altitude ~1.5-3.5 km [2] | |
| 09/03/2007 [a] | 41003 [*P2007a*] | 97 | 47-49 | Large plume (200 km) dispersing SE displaying significant uplift over the ocean (~2-6 km) [2] | |
| 10/05/2007 | 41469 [*P2007b*] | 97 | 47-48 | Large plume (220 km) dispersing SE at an altitude of ~2 km with pulses reaching 4 km [3] | 10 |
| 04/26/2009 [b] | 49755 [*P2009*] | 96 | 47-48 | Large plume (~100 km) identified at an altitude of ~1 km. Volcanic source outside MISR swath, source determined using Terra MODIS data. [2] | |
| 01/31/2011 | 59148 [*P2011a*] | 99 | 46-47 | Large plume (110 km) complex dispersion SE-NNE at an altitude of ~1.5 km with pulses up to ~4.5 km [1] | |
| 02/02/2011 | 59177 [*P2011b*] | 97 | 45-47 | Moderate plume (50 km) dispersing ENE at an altitude of ~3 km [N/A] | 15 |
| 02/18/2011 [a] | 59410 [*P2011c*] | 97 | 46-47 | Moderate plume (50 km) plume dispersing ENE at an altitude of ~2 km [1] | |
| 05/07/2011 | 60546 [*P2011d*] | 99 | 47-48 | Large plume (125 km) dispersing ESE displaying moderate uplift over the ocean ~2-4 km [2] | |
| 07/12/2011 | 61507 [*P2011e*] | 97 | 47-48 | Large multi-segment plume (80 km) dispersing E ranging in height ~3-8 km [2] | |
| 11/06/2013 | 73856 [*P2013*] | 97 | 47-48 | Large plume (180 km) dispersing SE at an altitude of ~1 km [1] | 20 |
| 02/27/2015 | 80817 [*P2015*] | 99 | 47-48 | Large bifurcated plume (90 km) dispersing ESE with the northern lobe at an altitude of ~3 km and the southern lobe at ~2 km. [1] | |
| 10/06/2016 [a,b] | 89365 [*P2016*] | 96 | 47-48 | Plume (50 km) identified at an altitude of ~1.5 km. Volcanic source outside MISR swath, source determined using Terra MODIS data. [3] | |

25  * Plume heights and dimensions based on preprocessing with MISR INteractive eXplorer (MINX) software.
[†] See Section 4.
[a] Plumes **not included** in the original Kamchatka eruption database in Flower & Kahn (2017a)
[b] Full MINX retrieval not possible. Heights and length based on a restricted retrieval.

**Table 2: MISR components from 774-mixture model identified in Karymsky plumes.**

| # | Model Code | r₁ (µm) | r₂ (µm) | rₑ (µm) | σ | SSA (446) | SSA (558) | SSA (672) | SSA (866) | Particle size/shape | Code[c] |
|---|---|---|---|---|---|---|---|---|---|---|---|
| 1 | sph_nonabs_ 0.06 | 0.002 | 0.329 | 0.056 | 1.650 | 1 | 1 | 1 | 1 | Small, Spherical, Non-Absorbing | SmSpNab ■ |
| 3 | sph_nonabs_ 0.26 | 0.005 | 1.690 | 0.262 | 1.750 | 1 | 1 | 1 | 1 | Small-Medium, Spherical, Non-Absorbing | MsSpNab ■ |
| 4 | sph_nonabs_ 0.57 | 0.008 | 3.805 | 0.568 | 1.800 | 1 | 1 | 1 | 1 | Medium, Spherical, Non-Absorbing | MeSpNab ■ |
| 5 | sph_nonabs_ 1.28 | 0.013 | 8.884 | 1.285 | 1.850 | 1 | 1 | 1 | 1 | Large, Spherical, Non-Absorbing | LaSpNab ■ |
| 6 | sph_abs_0.1 2_nr_1.50_0. 80_flat | 0.003 | 0.747 | 0.121 | 1.700 | 0.818 | 0.822 | 0.825 | 0.828 | Small Spherical, Highly Absorbing (Flat) | SmSpHab(f) ■ |
| 7 | sph_abs_0.1 2_nr_1.50_0. 80_steep | 0.003 | 0.747 | 0.121 | 1.700 | 0.838 | 0.822 | 0.801 | 0.756 | Small Spherical, Highly Absorbing (Steep) | SmSpHab(s) ■ |
| 8 | sph_abs_0.1 2_nr_1.50_0. 90_flat | 0.003 | 0.747 | 0.121 | 1.700 | 0.91 | 0.912 | 0.913 | 0.915 | Small, Spherical, Moderately Absorbing (Flat) | SmSpMab(f) ■ |
| 10[b] | dust_grains_ mode1 | 0.1 | 1 | 0.754 | 1.500 | 0.920 | 0.977 | 0.994 | 0.997 | Medium, Weakly-Absorbing Dust | MeNspWab ■ |

|  |  |  |  |  |  |  |  |  |  | Grains |  |
|---|---|---|---|---|---|---|---|---|---|---|---|
| 11[b] | spheroidal_ mode2 | 0.1 | 6 | 2.4 | 2.40 0 | 0.81 | 0.902 | 0.971 | 0.983 | Large, Weakly-Absorbing Coarse Dust Spheroids | LaSpdWab ■ |

[a] Each model is comprised of an amalgamation of particles with a designated effective radius ($r_e$), ranging between a minimum ($r_1$) and maximum ($r_2$) particle size. Full model details available in *Limbacher & Kahn* (2014).

[b] Dust and spheriods optical properties from *Kalashnikova et al.* (2005)

[c] Code for each component incorporating three elements: **Size** – Small (Sm), Medium (Md), Large (La), Very Large (VLa); **Shape** – Spherical (Sp), Spheroid (Spd), Non-spherical (Nsp); **Absorption** – Non-Absorbing (Nab), Weakly Absorbing (Wab), Moderately Absorbing (Mab), Highly Absorbing (Hab); **Spectral Absorption Profile** – Equal ('flat') absorption in all spectral bands (f); Varying ('steep') absorption across spectral bands (s

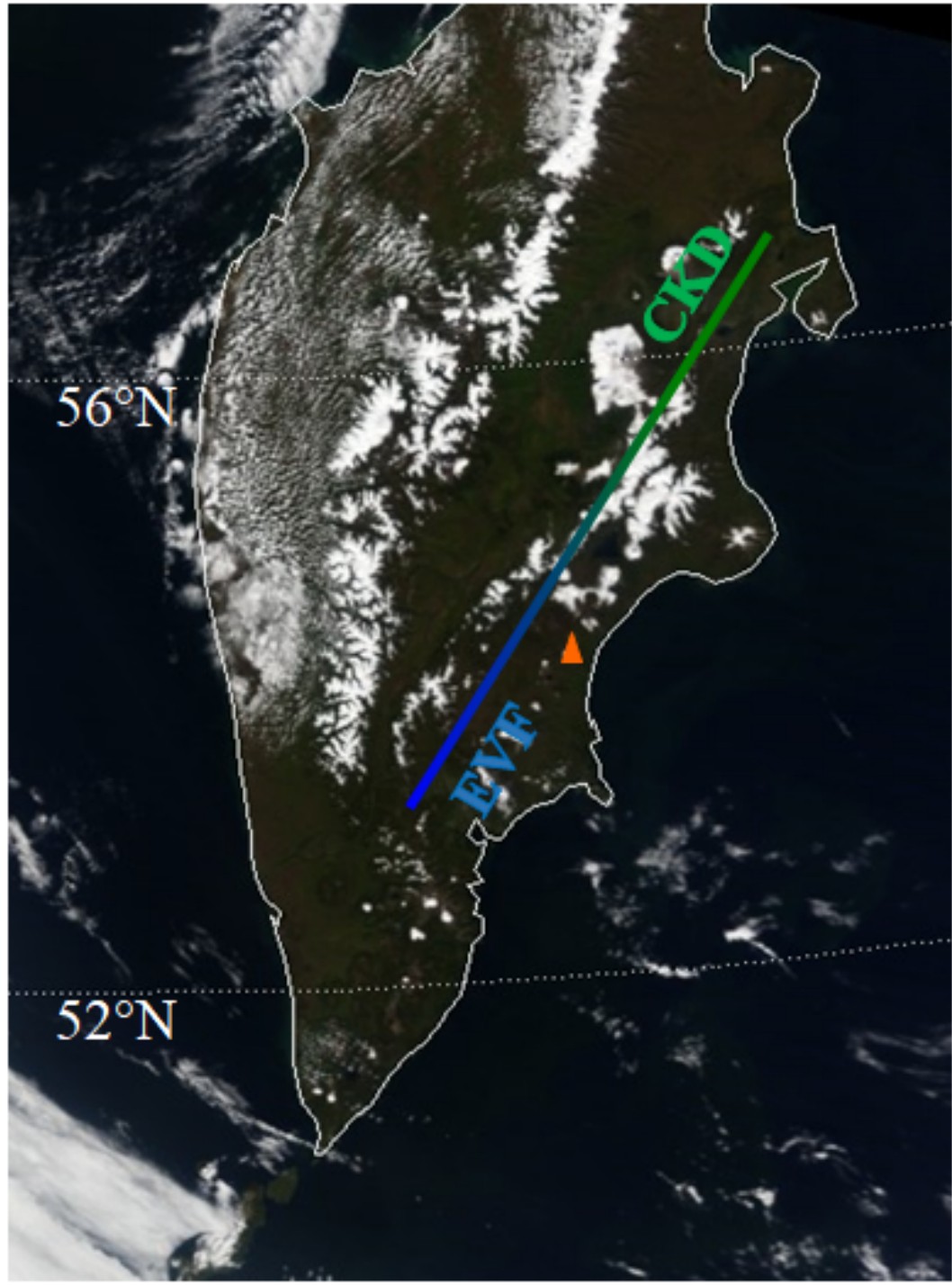

**Figure 1: MODIS image of the Kamchatka Peninsula (Russia) on September 22, 2016. The orange triangle denotes the location of Karymsky volcano. The dominant geological features dividing the Kamchatka Peninsula, the Central Kamchatka Depression (CKD) and Eastern Volcanic Front (EVF), are also labeled.**

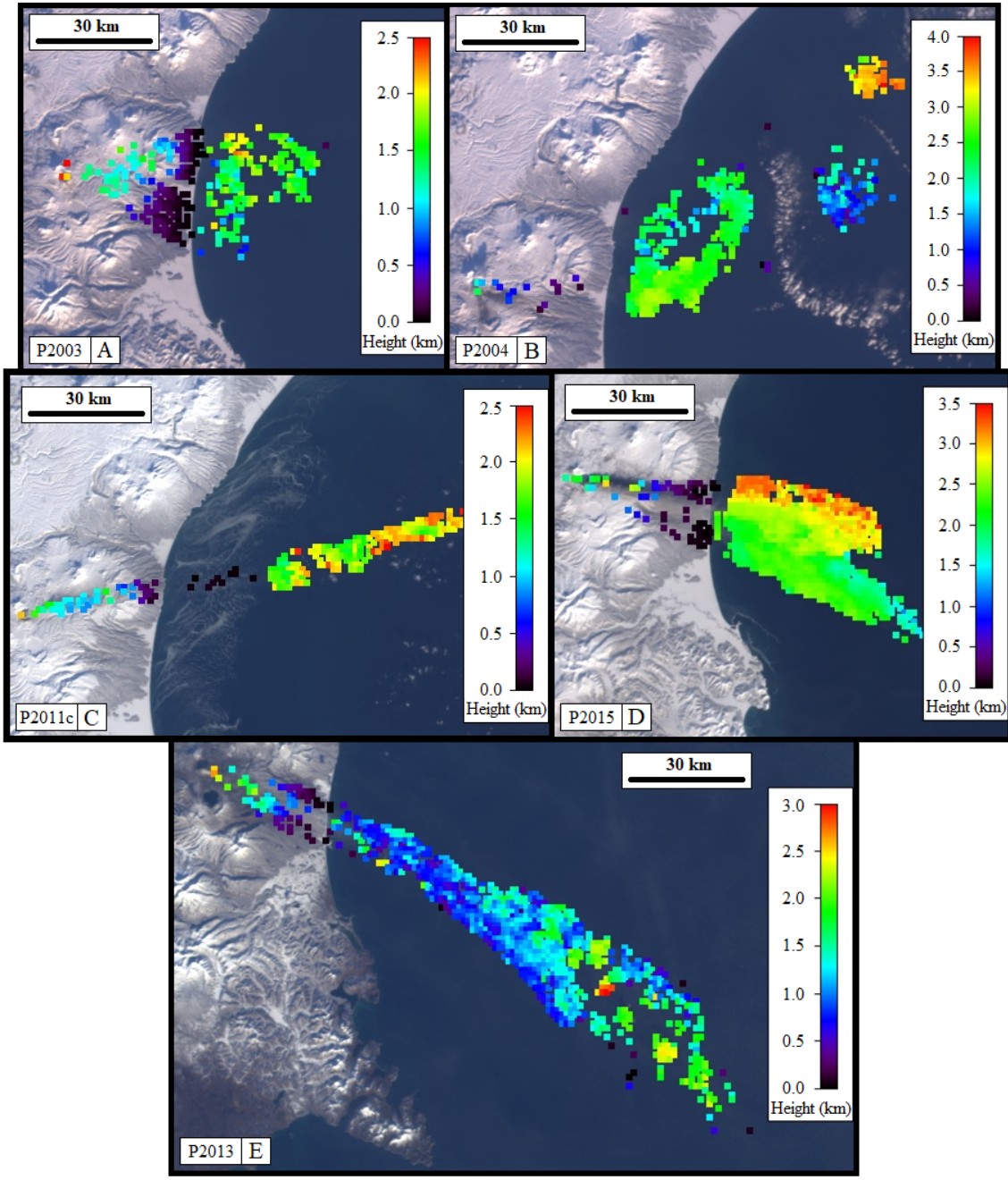

**Figure 2: MISR stereo height retrieval maps derived using the MISR INteractive eXplorer (MINX). Class 1 plumes (low MeNspWab) emitted from Karymsky volcano on: a) December 27, 2003 (Orbit: 21402; Path: 99; Block: 47); b) January 12, 2004 (Orbit: 21635; Path: 99; Blocks: 46-47); c) February 27, 2015 (Orbit: 80817; Path: 99; Blocks: 47-48); d) February 18, 2011 (Orbit: 59410; Path: 97; Block: 46-47) and e) November 6, 2013 (Orbit: 73856; Path: 97; Blocks: 47-48). Note that the elevation scales vary from panel to panel.**

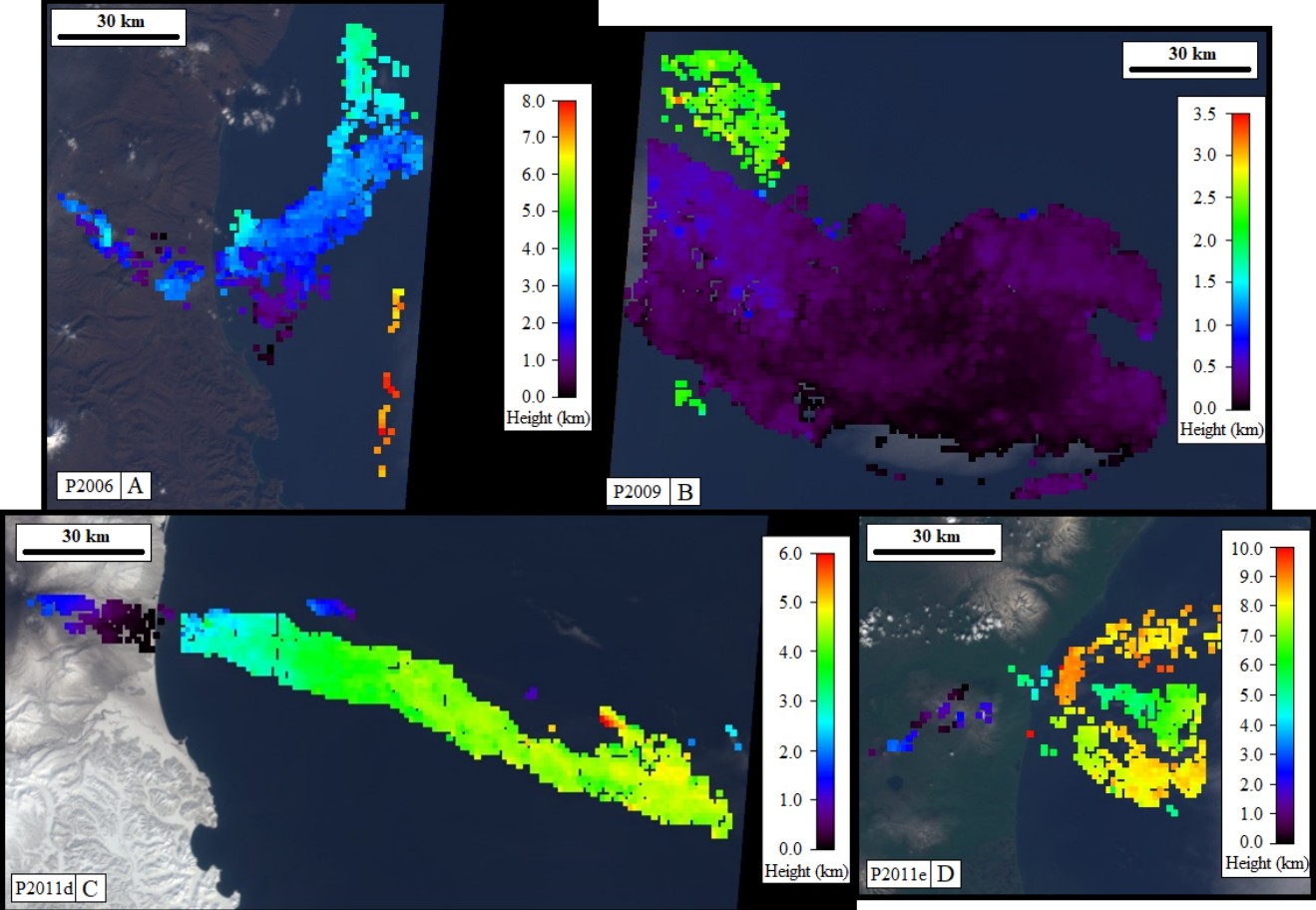

**Figure 3: MISR stereo height retrieval maps derived using the MISR INteractive eXplorer (MINX). Class 2 plumes (moderate MeNspWab) emitted from Karymsky volcano on: a) October 23, 2006 (Orbit: 36416; Path: 97; Blocks: 47-48); b) July 12, 2011 (Orbit: 61507; Path: 97; Blocks: 47-48); c) May 7, 2011 (Orbit: 60546; Path: 99; Blocks: 47-48); and d) April 26, 2009 (Orbit: 49755; Path: 96; Blocks: 48-49);. Note that the elevation scales vary from panel to panel.**

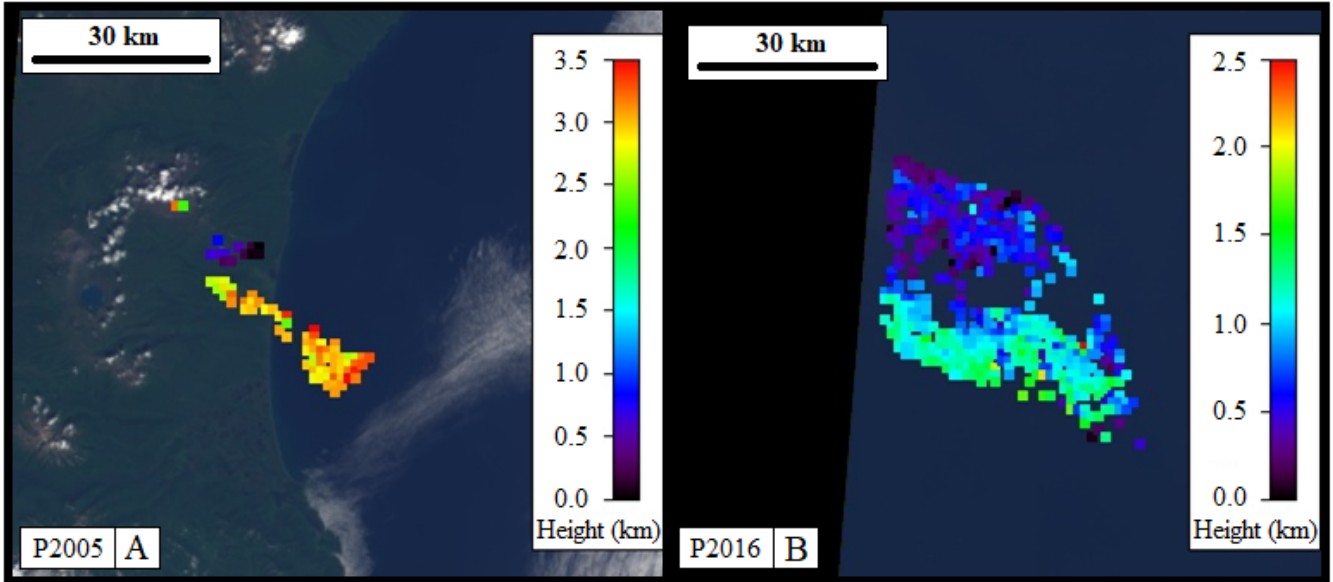

**Figure 4: MISR stereo height retrieval maps derived using the MISR INteractive eXplorer (MINX). Class 3 plumes (high MeNspWab) emitted from Karymsky volcano on: a) August 28, 2005 (Orbit: 30285; Path: 97; Block: 47); and b) October 6, 2016 (Orbit: 89365; Path: 96; Blocks: 47-48). Note that the elevation scales vary from panel to panel.**

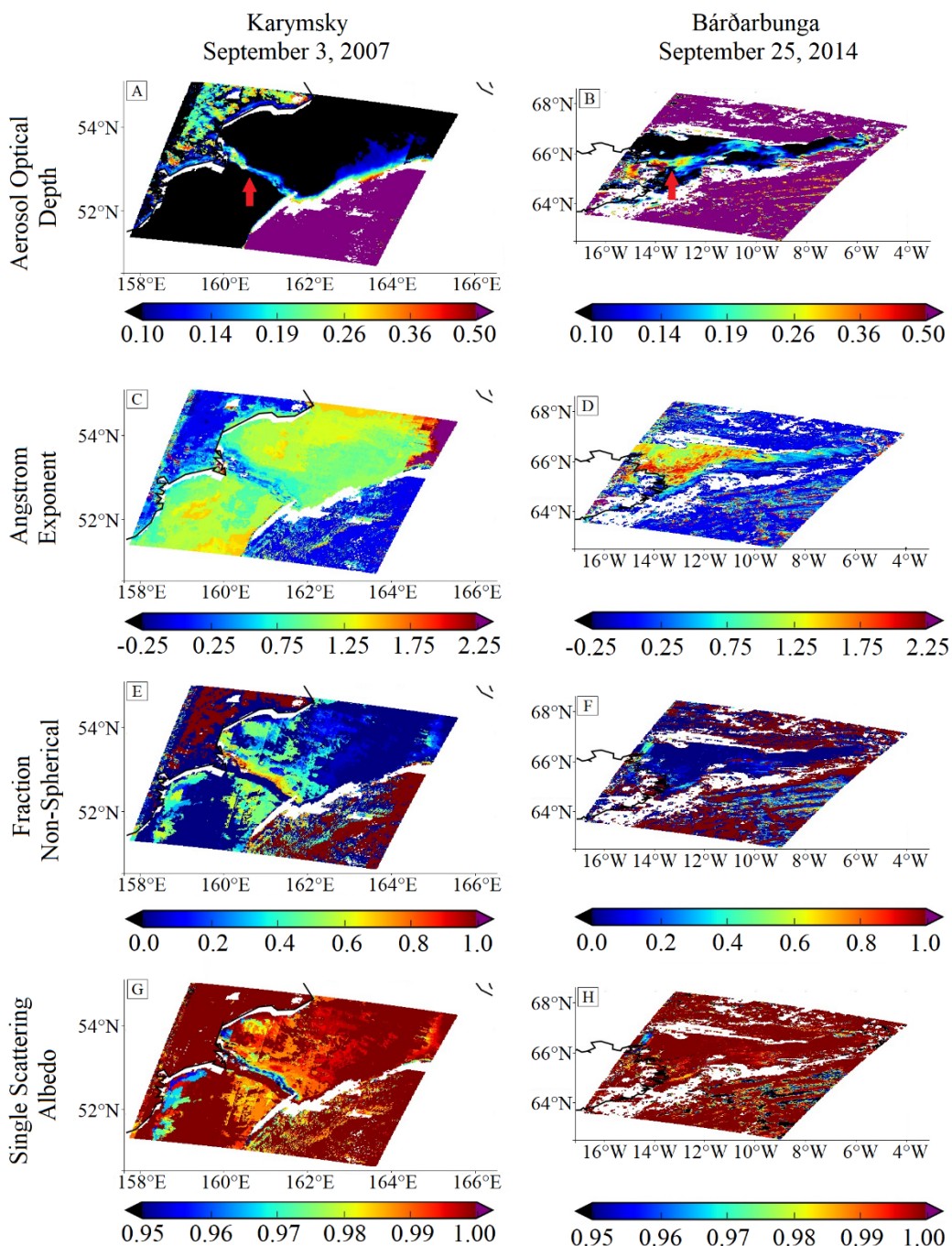

**Figure 5: MISR Research Aerosol (RA) algorithm retrievals of: Aerosol Optical Depth (A & B), Angstrom Exponent (C & D), Fraction of Non-Spherical particles (E & F) and Single Scattering Albedo (G & H) for a plume emitted from Karymsky volcano on September 3, 2007 (Orbit: 41003; Path: 97; Blocks: 47-48); and an SO$_2$ rich plume from Bárðarbunga volcano, Iceland, emitted on September 25, 2014 (Orbit: 78567; Path: 214; Blocks: 37-38). Red arrows indicate plume locations in panels A and B.**

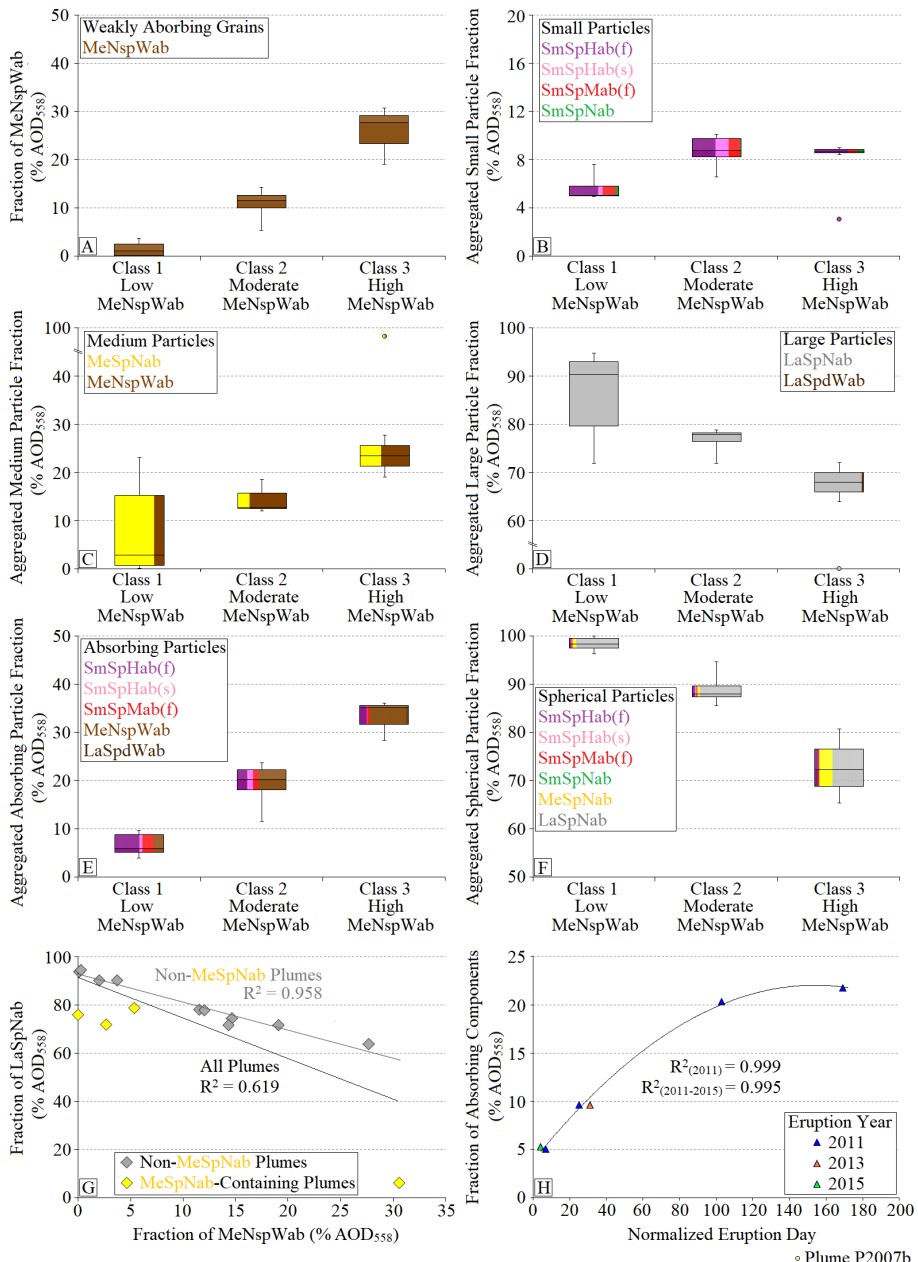

**Figure 6: Box plots of the relative proportions (in quartiles, with the mean represented by a line within the box, dot in Class 3 (b-d) represents the outlying results of *P2007b*) of: a) Medium, non-spherical, weakly absorbing (MeSpNab) used to define the three plume classes; b) Small particles (including: SmNspNab, SmSpHab(f), SmSpHab(s), SmSpMab(f)); c) Medium particles (including: MeSpNab and MeNspWab); d) Large particles (including: LaSpNab and LaSpdWab); e) absorptivity of plumes; f) relative sphericity, all stratified by Class type; g) Fraction of large spherical particles (LaSpNab) relative to medium, grains (MeNspWab) for each Karymsky plume, including correlation statistics for all samples and non-MeSpWab containing plumes; and h) Fraction of absorbing components relative to normalized eruption day for plumes observed from 2011 to 2015. Note: Scales differ between components. The color fraction of each box plot (as defined in Table 2 and earlier figures) indicates the relative proportion of each contributing aerosol component.**

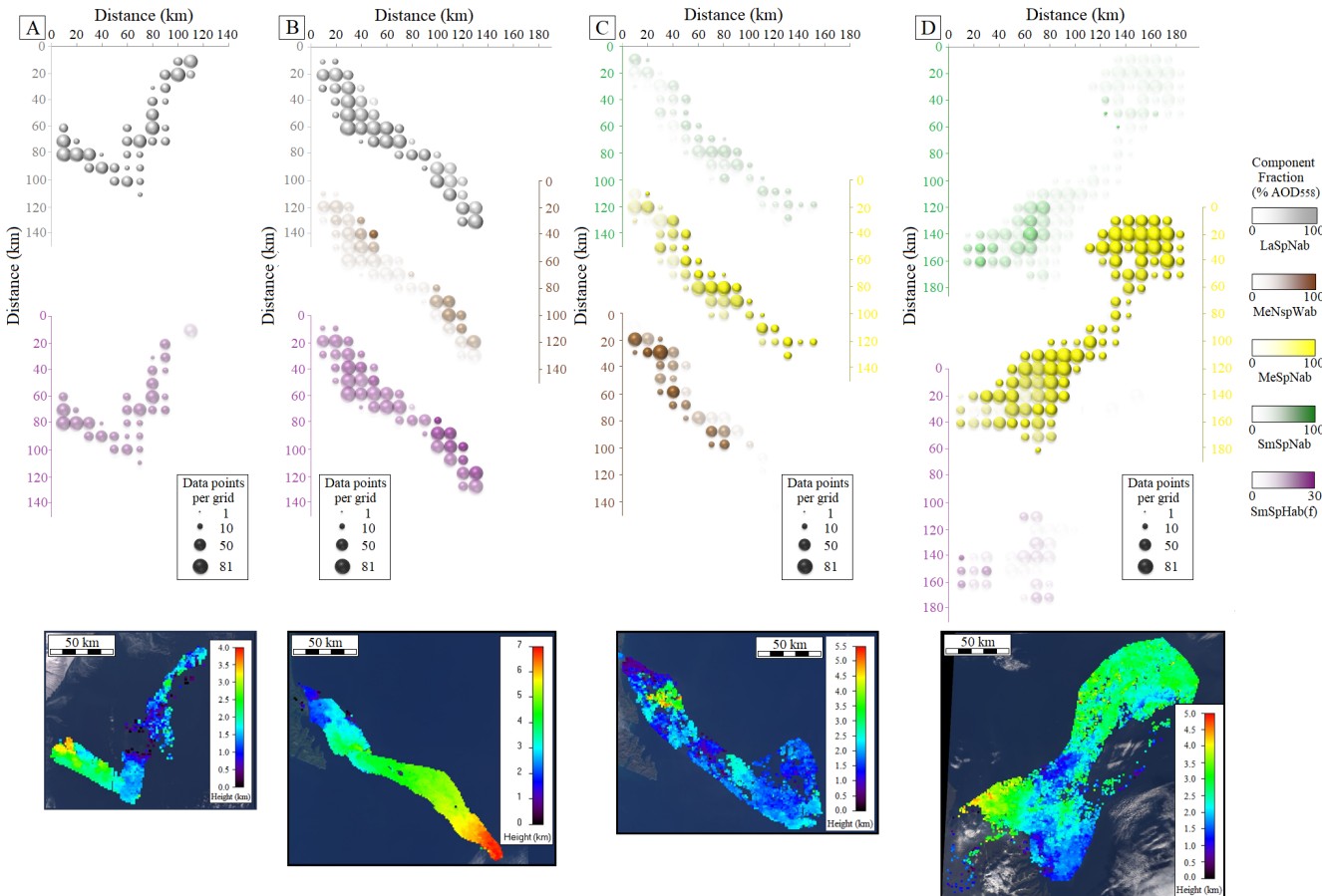

**Figure 7: Analysis of MISR Research Aerosol (RA) algorithm retrieved particle properties (for over-ocean plume regions), aggregated to a ~10 km grid, with corresponding MISR stereo height retrieval maps derived using MISR INteractive eXplorer (MINX). In the upper panels, the degree of dot color saturation indicates the percent mid-visible AOD of that component, and the dot size represents the number of 1.1 km retrievals within the 10 km grid cell for which that component was present in the retrieval results. Plumes were emitted from Karymsky volcano on: a) January 31, 2011 (Orbit: 59148; Path: 99; Block: 47); b) September 3, 2007 (Orbit: 41003; Path: 97; Blocks: 47-48); c) October 5, 2007 (Orbit: 41469; Path: 97; Blocks: 47-48); and d) Bárðarbunga volcano, Iceland, emitted on September 25, 2014 (Orbit: 78567; Path: 214; Blocks: 37-38). Particle types: Small, spherical, highly absorbing (SmSpHab(f)); small, spherical, non-absorbing (SmSpNab); medium, spherical, non-absorbing (MeSpNab); large, spherical, non-absorbing (LaSpNab); medium, non-absorbing dust grains (MdNspWab). Note: The elevation scales vary between the panels, and particle concentrations scales vary by component type.**

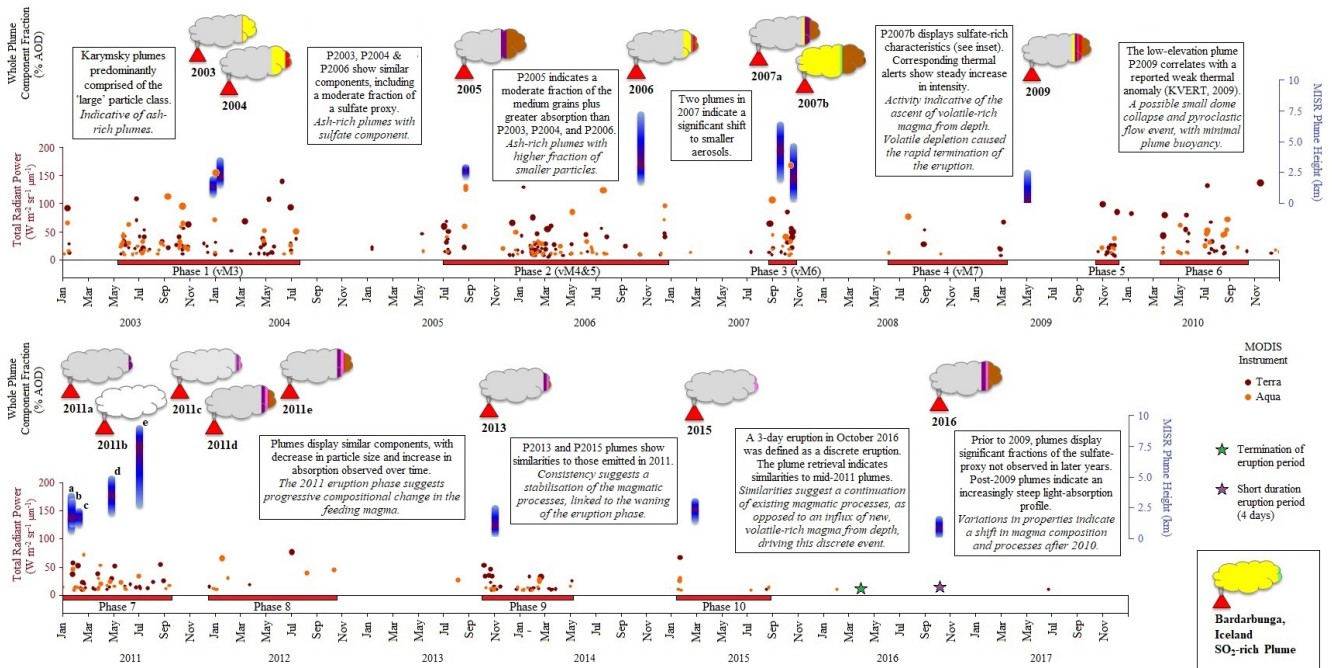

**Figure 8:** Timeline of volcanic activity at Karymsky volcano. [Top row of each panel] MISR Research Aerosol Retrieval (RA) algorithm plume particle component fractions for individual plumes (the colors are identified in Table 2 and the legend of Figure 3; White indicates no data available). [Middle row and right axis] MISR stereo-derived plume heights, indicating exact timing and vertical extent (blue) and vertical distribution of highest contrast elements (purple) for each plume. [Bottom row and left axis] Thermal anomalies obtained from the MODVOLC hotspot alerts from both the Terra and Aqua MODIS satellite instruments. [Horizontal axis] Eruption phases defined by MODIS thermal alerts and correlated with phases (vM) defined in the work of van Manen et al., (2012). [Inset] Plume properties for Bárðarbunga plume on September 25, 2014 (Orbit: 78567; Path: 214; Blocks: 37-38).