# Peer review of "Tracking microphysical variations in emissions from Karymsky volcano using MISR multi-angle imagery, and implications for volcanological interpretation"

_Atmospheric Chemistry and Physics, 2017_

## Referee Comment (RC1) · C. Hayer (Referee) · 28 Sep 2017

**General comments**

This manuscript uses the MISR instrument, in combination with thermal alerts from MODIS, to investigate the variation of the composition of plumes emitted from Karymsky volcano over the lifetime of MISR (2000 – 2017). In spite of limitations in the MISR retrieval algorithm and the constraints at the upper end of the spectral range of the instrument, the authors are able to retrieve qualitative variations within a single plume over its horizontal extent as well as between plumes over time. From this, variations in

the eruptive style and regime are deduced. The authors were able to demonstrate the ability of MISR to clearly distinguish between plumes dominated by sulfate particles and those dominated by ash particles.

Overall, this is an interesting article and the science within seems sound. I support the paper for publication once the concerns below are addressed.

**Specific comments**

P9 Para3 (beginning L18): The authors discuss an eruption observed from ground-based instruments by Lopez et al. (2015) as a way to verify the processes inferred by the satellite-based observations. However, there doesn't appear to have been a MISR observation to compare the Lopez et al. measurements to and so this seems to be mostly speculation on the part of the authors as to what MISR *might* have been. I acknowledge that ground truthing observations from any satellite instrument is hard and made harder by the narrow swath of the MISR instrument but I'm not sure that this comparison can be drawn. If it is included, I think the authors need to be clearer about this being speculative.

Conclusions (P12-13): The authors ascribe the differences in the activity and plume composition from the volcano pre- and post-2010 as the end and beginning of a cycle. I am not convinced, by the data shown, that there is sufficient evidence that this is a long-term cycle rather than a change/evolution of the magmatic regime. I am not ruling out the possibility of this being a long-term cycle, simply that I don't feel the current data is sufficient to determine either way. There does appear to be cyclicity, as the authors say, especially post-2010. The support for this cyclicity I feel is quite strong, with the similar variations in composition shown in Fig. 6h when the measurements are normalized for eruption day. The lack of this shorter-term cyclicity and the change in the plume compositions retrieved pre- and post-2010 could equally suggest a change in the regime, rather than a different part of a longer-term cycle. Could the authors either present the data that led to their conclusion of a long-term cycle or rewrite this part of the conclusion.

[Figure]

**Technical corrections**

P5 L3: "SSA" not defined

P5 L14: 10 km$^2$ grids (add square)

P5 L20-21: Table 2 has two small spherical absorbing flat profiles listed, highly moderately. Do the authors mean the combination of these? Could this be made a little clearer.

P7 L23: The figures displaying variations in the small, medium & large particles are Fig. 6 b, c & d rather than just Fig. 6c as written.

P12 L25-29: This is all one sentence. Please rewrite to reduce the length/split into more sentences.

P13 L4: The sentence reads as plural but only one Bardarbunga eruption plume was considered.

Table 2, title row: What do the "r"s refer to? I am assuming re is effective radius but r1 and r2? Could these be explained in the footnotes.

Table 2, column 1: s 2 and 9 missing from the table - is this intentional?

Fig. 1: This figure is not referenced in the main text.

Figs. 2, 3 & 4: Lat-lon markers or a length scale (preferably) should be included in these figures. The authors discuss the variable horizontal extent of the plumes on page 6 (L6-7) but the reader cannot distinguish this for themselves.

Figs. 2 and 3 b & c: There appears to be an abrupt increase in the plume height retrieved when the plume moves over the water. Is this real or an artifact of the RA? It seems to happen to all of the plumes shown in Fig. 2 and those in Fig. 3 b & c (though not Fig. 3 a). There is a significant variation in the plume shown in Fig. 3 d but there does not seem to be any coastline that may have caused it. The same sudden change is also seen in a number of the plumes in the supplementary material. Could the authors please elaborate on possible causes of these variations?

Fig. 6 h: In the text, the authors include the $R^2$ value for 2011 only as well as for all of the 2011-2015 plumes. It may be worth adding the $R^2$ value for just 2011 to the plot as well as the full dataset $R^2$?

Fig. 7 top: The 3D-effect used on the plot, while aesthetically pleasing, makes distinguishing the saturation of the point much more difficult, especially on the grey data set (LaSpNab).

Fig. 7 legend: In the text (Page 8, L14-15), the authors describe the 2007b plume (panel C) as being dominated by the sulfate proxy (MeSpNab, yellow) and medium grains (MeNspWab). The latter dataset is not denoted on the legend. I'm not certain if the data set is missing or mislabeled – it could be that the brown data set should be this data set – currently labeled MeNspNab. Could it either be included or labeled correctly.

Fig. 7 bottom: Could the wind direction be shown on the plots so that the variation over the plume described in the text can be more easily identified?

---

## Referee Comment (RC2) · Anonymous Referee #2 · 27 Oct 2017

General Comments

The paper presents MISR data that are used, in combination with the MODIS thermal alerts, to investigate volcanic plumes from Karymsky volcano between 2000 and 2017. MISR retrievals showed 2-4 km high volcanic plumes having different particle types. From their analysis, the authors identified two eruptive cycles, producing plumes mainly composed by sulfate prior to 2010 and plumes with varying fraction of absorption particles after 2011. The analysis of MISR data is well written but, before the publication, the volcanological interpretation should be improved and validated with other data.

Specific points

1. A detail description of volcanic setting and eruptions for the Karymsky volcano is lacking. The authors should cite other papers and go in depth on some works that could improve their understanding on typical volcanic activity (e.g. Johnson et al.,1998; Ozerov et al., 2003). In the 1.1. Karymsky volcano chapter, the authors should add a table which includes the eruptive events with their main features.

2. Volcanological interpretation is not well supported. Authors should improve the eruption description during MISR observations in chapter 4. It is not very clear what is the eruption style that MISR is detecting (e.g. ash emission, Strombolian activity, lava flow emissions). Although MISR is able to distinguish volcanic ash plumes from degassing/water vapor plumes, data shown in the paper are not able to define the eruption style. Furthermore, the classification among Strombolian or Vulcanian eruptions cannot done only on the base of the column height or duration of the eruptive event. Finally, the differences among the eruption periods should be again supported by other data, mainly due to the small coverage of the MISR sensor (about 4%).

3. Authors state that MISR was able to detect particle fallout, physical aggregation, chemical evolution only qualitatively but they should clarify how they reached this objective point to point. To validate MISR data and their interpretations they could improve the analysis of MODIS data applying well known algorithms as the Brightness Temperature Difference (BTD) technique (Wen and Rose, 1994).

4. Volcanic plumes are strongly affect by atmospheric fields. The maximum distance reached by volcanic particles, for example, could depend on the wind speed and, mainly for this reason, I suggest to insert a new figure which includes wind profiles for each event retrieved by MISR.

Technical corrections

Replace the title 'Tracking microphysical variations in emissions from Karymsky volcano using MISR multi-angle imagery, and implications for volcano geologic interpretation' with 'Tracking microphysical variations in emissions from Karymsky volcano using MISR multi-angle imagery, and implications for volcanological interpretation'

P1L15. Clarify the sentence. What do the authors mean for 'high volcanic surface manifestation'?

P1L16. Add the size.

P1L22. See specific points about the interpretation in terms of activity cycles.

P3L10. The authors state that MISR has the potential to distinguish the emission from (a) ash explosions, (b) pulsatory degassing, (c) gas jetting, and (d) explosive activity. May the authors identify those emissions for each MISR data shown in this paper?

P3L12-13. Delete this sentence. See specific points.

P3L18-20. This is in general true at the same atmospheric conditions. I suggest to delete this sentence.

P4L17. Explain 'derive proxy particles type'.

P5L3. Define SSA.

P5L13. Add the period.

P5L15. Add how the size, shape and absorption are retrieved by the MISR RA.

P5L20. Add the size retrieved by MISR for large, medium and small particles.

P8L15-17. Due the very few MISR observation rate (about 4%), this sentence is not well supported.

P9L14. Add the distance from the volcanic vent.

P10. Replace 'geological' with 'volcanological' in the sub-chapter 4.2.

P10L10. Improve the description of the eruptive phases.

P10L24. I wonder if the shift from effusive to explosive activity is given only by the amount of ash in the atmosphere without taking into account the amount of sulfate/water that could be also high for both activities. May the authors add some references?

P11L2-5. Clarify this sentence.

P11L11. Fig. 4c is lacking.

P11L18. The hypothesis of pyroclastic flows should be justified by published papers or news from web-sites.

P11L26. Is MISR able to distinguish among sulfate or water vapor plumes?

P12L16-17. Clarify.

P12L21-22. Clarify.

P12L25. What is 'volcano's geologic evolution' for?

P12L28. The eruption style cannot be derived from MISR data analysis. See specific points.

P12L32. Add the classes.

P13L10-11. I wonder if this analysis is affected by the small percentage of MISR data respect to the eruptions happened in the same period.

P13L14. See specific comments on eruptive cycles.

P13L18-19. Are MISR data able to discriminate those processes? How?

References

Johnson, J.B., Lees, J.M., Gordeev, E.I. (1998), Degassing Explosions at Karymsky Volcano, Kamchatka, Geophysical Research Letters, DOI: 10.1029/1998GL900102. Ozerov, A., Ispolatov, I., Lees, J.(2003), Modeling Strombolian eruptions of Karymsky

volcano, Kamchatka, Russia, Journal of Volcanology and Geothermal Research 122, 265-280. Wen, S., and W. I. Rose (1994), Retrieval of sizes and total masses of particles in volcanic clouds using AVHRR bands 4 and 5, J. Geophys. Res., 99, 5421–5431, doi:10.1029/93JD03340.

Figures

Figure 1 is not cited in the text.

Figure 2, 3 and 4. Add the scale in km.

Figure 7. Add the scale at the bottom of figures. In Fig. 7 B, the plume height of 7 km reached far from the volcanic vent is higher than the height above the vent. Why?

Table 1. Improve the eruption description.

I suggest to move the plots from the supplementary material to the paper and, moreover, add the plots 'height versus distance from the volcanic vent' as retrieved by the MINX software. Finally, the column heights reported in the paper should be above sea level.

---

## Author Comment (AC1) · 6 Dec 2017

Response to reviewer 1 - C. Hayer (Referee)

ˆ Indicates reviewer comment *** Indicates response from authors. All page (P) and line (L) references relate to the track changes version of the manuscript attached as a supplement.

Specific comments

ˆP9 Para3 (beginning L18): The authors discuss an eruption observed from ground-based instruments by Lopez et al. (2015) as a way to verify the processes inferred by the satellite-based observations. However, there doesn't appear to have been a MISR observation to compare the Lopez et al. measurements to and so this seems to be mostly speculation on the part of the authors as to what MISR might have been. I acknowledge that ground truthing observations from any satellite instrument is hard and made harder by the narrow swath of the MISR instrument but I'm not sure that this comparison can be drawn. If it is included, I think the authors need to be clearer about this being speculative.

***We agree with the reviewer's frustration but unfortunately at Karymsky there were no coincident ground-based or in situ observations with MISR overpasses. The sections mentioned here (P4L12-26) have been updated to better outline that the observations mentioned are included as a general reference and not for direct validation purposes. We feel it is important to present such comparisons, as they represent the best that can be offered in the present situation. Note also that Reviewer 2 actually asked for more discussion of other (necessarily non-coincident) Karymsky observations. We hope to undertake direct comparisons to ground-based and in situ observations as we extend this technique to other volcanoes with more extensive observations (e.g. Iceland).

ˆConclusions (P12-13): The authors ascribe the differences in the activity and plume composition from the volcano pre- and post-2010 as the end and beginning of a cycle. I am not convinced, by the data shown, that there is sufficient evidence that this is a long-term cycle rather than a change/evolution of the magmatic regime. I am not ruling out the possibility of this being a long-term cycle, simply that I don't feel the current data is sufficient to determine either way. There does appear to be cyclicity, as the authors say, especially post-2010. The support for this cyclicity I feel is quite strong, with the similar variations in composition shown in Fig. 6h when the measurements are normalized for eruption day. The lack of this shorter-term cyclicity and the change in the plume compositions retrieved pre- and post-2010 could equally suggest a change

in the regime, rather than a different part of a longer-term cycle. Could the authors either present the data that led to their conclusion of a long-term cycle or rewrite this part of the conclusion.

***Thank you, for this comment. We have updated the conclusions (beginning P20L1) to better clarify that, although we observe ongoing evolution in the 1996-2016 eruption (2000-2016 observations), there is also an element of cyclicity in the formation eruptive phases at Karymsky.

Technical corrections

^P5 L3: "SSA" not defined

***P7L9 – Thank you, we have added the definition. SSA = Single-scattering albedo

^P5 L14: 10 km2 grids (add square)

***P7L23 – Thank you, this change has been made.

^P5 L20-21: Table 2 has two small spherical absorbing flat profiles listed, highly moderately. Do the authors mean the combination of these? Could this be made a little clearer.

***P7L8-14 – One of the flat profiles displays strong absorption SSA ~0.8 (highly absorbing), the other displays less absorption SSA ~0.9 (moderately absorbing). The SSA values are included in the table, and the section in the paper detailing these differences has been clarified.

^P7 L23: The figures displaying variations in the small, medium & large particles are Fig. 6 b, c & d rather than just Fig. 6c as written.

***P11L6 – Thank you, this change has been made.

^P12 L25-29: This is all one sentence. Please rewrite to reduce the length/split into more sentences.

***P19L7-13 – Thank you, this change has been made.

^P13 L4: The sentence reads as plural but only one Bardarbunga eruption plume was considered.

***P17L10-18 – Thank you, this statement and earlier sections has been updated to indicate that although only one plume is shown in Fig 5 & 7, two additional Bardarbunga plumes were analysed and the details are now included in the supplemental table.

^Table 2, title row: What do the "r"s refer to? I am assuming re is effective radius but r1 and r2? Could these be explained in the footnotes.

***P26L1 – Thank you, this change has been made.

^Table 2, column 1: s 2 and 9 missing from the table - is this intentional?

***P25L1 – Thank you, for you comment. These are intentionally missing as the MISR RA components 2, 9, 11, 12 and 13 were not retrieved in Karymsky plumes and therefore are not relevant here. The header text indicates that only components identified in Karymsky plumes were included.

^Fig. 1: This figure is not referenced in the main text.

***P3L19 – Thank you, this change has been made.

^Figs. 2, 3 & 4: Lat-lon markers or a length scale (preferably) should be included in these figures. The authors discuss the variable horizontal extent of the plumes on page 6 (L6-7) but the reader cannot distinguish this for themselves.

***P28-30 – Thank you, this change has been made.

^Figs. 2 and 3 b & c: There appears to be an abrupt increase in the plume height retrieved when the plume moves over the water. Is this real or an artifact of the RA? It seems to happen to all of the plumes shown in Fig. 2 and those in Fig. 3b&c (though not Fig. 3 a). There is a significant variation in the plume shown in Fig. 3 d but

there does not seem to be any coastline that may have caused it. The same sudden change is also seen in a number of the plumes in the supplementary material. Could the authors please elaborate on possible causes of these variations?

***P28-30 – Thank you, for your comment. The decreasing plume height toward the coast followed by sharp uplift is a function of particles being contained within the lower atmosphere. As air transitions over water the lower atmosphere is subjected to less friction forces causing an increase in wind speed (plots added to supplemental figures) and plume uplift. Additional details and references (particularly Flower & Kahn, GRL 2017b, that discusses this phenomenon in detail) have been added to the manuscript (P9L9-13). In some cases ( e.g P2007b) significant plume uplift is driven by local frontal systems, details have been added to the text (P9L9-13).

^Fig. 6 h: In the text, the authors include the R2 value for 2011 only as well as for all of the 2011-2015 plumes. It may be worth adding the R2 value for just 2011 to the plot as well as the full dataset R2?

***P32– Thank you, this change has been made.

^Fig. 7 top: The 3D-effect used on the plot, while aesthetically pleasing, makes distin-guishing the saturation of the point much more difficult, especially on the grey data set (LaSpNab).

***P33 – Thank you, this change has been made. The shadow has been removed from these figures to improve the variations in color density more easy to discern.

^Fig. 7 legend: In the text (Page 8, L14-15), the authors describe the 2007b plume (panel C) as being dominated by the sulfate proxy (MeSpNab, yellow) and medium grains (MeNspWab). The latter dataset is not denoted on the legend. I'm not certain if the data set is missing or mislabeled – it could be that the brown data set should be this data set – currently labeled MeNspNab. Could it either be included or labeled correctly.

***P33 – Thank you, this change has been made.

^Fig. 7 bottom: Could the wind direction be shown on the plots so that the variation over the plume described in the text can be more easily identified?

***Supplemental – Thank you, wind plots have been added to the Supplemental Data along with plume profile plots

Please also note the supplement to this comment:
https://www.atmos-chem-phys-discuss.net/acp-2017-868/acp-2017-868-AC1-supplement.pdf

---

## Author Comment (AC2) · 6 Dec 2017

Response to Anonymous Referee #2

ˆ Indicates reviewer comment *** Indicates response from authors. All page (P) and line (L) references relate to the track changes version of the manuscript attached as a supplement.

General Comments

ˆThe paper presents MISR data that are used, in combination with the MODIS thermal alerts, to investigate volcanic plumes from Karymsky volcano between 2000 and 2017. MISR retrievals showed 2-4 km high volcanic plumes having different particle types. From their analysis, the authors identified two eruptive cycles, producing plumes mainly composed by sulfate prior to 2010 and plumes with varying fraction of absorption particles after 2011. The analysis of MISR data is well written but, before the publication, the volcanological interpretation should be improved and validated with other data.

***All line references relate to the track changes version of the manuscript.

Specific points

ˆ1. A detail description of volcanic setting and eruptions for the Karymsky volcano is lacking. The authors should cite other papers and go in depth on some works that could improve their understanding on typical volcanic activity (e.g. Johnson et al.,1998; Ozerov et al., 2003). In the 1.1. Karymsky volcano chapter, the authors should add a table which includes the eruptive events with their main features.

***P3L19-P4L26 – Thank you, this section has been updated to include more details about the volcano, its geological setting and eruption characteristics. Additionally, details of eruptions during the MISR observation period (2000-2017) compiled from the GVP have been included in the Supplemental Material.

ˆ2. Volcanological interpretation is not well supported. Authors should improve the eruption description during MISR observations in chapter 4. It is not very clear what is the eruption style that MISR is detecting (e.g. ash emission, Strombolian activity, lava flow emissions). Although MISR is able to distinguish volcanic ash plumes from degassing/water vapor plumes, data shown in the paper are not able to define the eruption style. Furthermore, the classification among Strombolian or Vulcanian eruptions cannot done only on the base of the column height or duration of the eruptive event. Finally, the differences among the eruption periods should be again supported by other

data, mainly due to the small coverage of the MISR sensor (about 4%).

***Thank you for your comment. The section discussing Karymsky eruption types (P4L11-26) has been edited to clarify that we are only analyzing the largest (ash explosions/Vulcanian) eruptions. This paper was never intended to imply we could discern eruption type (Strombolian/Vulcanian) from MISR images alone. The information on eruption types was included to acknowledge that we are only dealing with a small subset of the plumes from Karymsky (P4L23-26). The eruptions phases (P15L1-8) were not defined by the MISR plume data but by the MODIS (Terra and Aqua thermal anomaly record; which includes ∼4 observations in Kamchatka per day), in combination with published studies on complementary investigations with AVHRR (van Manen et al., 2012).

ˆ3. Authors state that MISR was able to detect particle fallout, physical aggregation, chemical evolution only qualitatively but they should clarify how they reached this objective point to point. To validate MISR data and their interpretations they could improve the analysis of MODIS data applying well known algorithms as the Brightness Temperature Difference (BTD) technique (Wen and Rose, 1994).

***Thank you for your comment. Details of the expected characteristics of each process are located in section 4.1 at the following line references: particle fallout (P13L8-11), physical aggregation (P13L11-13) and chemical evolution (P13L13-20). The plumes that display each characteristic are also included in this section. References to the Supplemental Data have been added to point to the graphical evidence from our analysis of each process, as these are too large/extensive to include in the main paper. The BTD techniques suggested, use thermal channels for retrievals and therefore track larger particles (>∼ 2 microns) than the ones MISR is most sensitive to; the MISR-observed particles are likely to stay aloft longer, especially if gravitational settling is the main removal mechanism. In respect to plume heights, Flower & Kahn (2017a; JVGR) compared MISR heights and those obtained from BTD techniques. This comparison indicated that BTD heights fell within the observed MISR height range. However, BTD

heights were skewed below the region of highest spatial contrast identified in the MISR height retrievals, suggesting that the BTD heights have a tendency to underestimate actual plume height.

ˆ4. Volcanic plumes are strongly affect by atmospheric fields. The maximum distance reached by volcanic particles, for example, could depend on the wind speed and, mainly for this reason, I suggest to insert a new figure which includes wind profiles for each event retrieved by MISR.

***Supplement – Thank you for your comment. MISR wind field retrievals have been added to the supplemental data sheets for each plume and are referenced in the text.

Technical corrections

ˆReplace the title 'Tracking microphysical variations in emissions from Karymsky volcano using MISR multi-angle imagery, and implications for volcano geologic interpretation' with 'Tracking microphysical variations in emissions from Karymsky volcano using MISR multi-angle imagery, and implications for volcanological interpretation'

***P1L1 – Thank you, this change has been made.

ˆP1L15. Clarify the sentence. What do the authors mean for 'high volcanic surface manifestation'?

***P1L14-17 – This section has been revised for clarity. By 'high volcanic surface manifestation' we mean "periods of time when lava flows and other radiating features are prevalent at the volcano, causing a high number of observations from the incorporated instruments".

ˆP1L16. Add the size.

***P1L18 – Thank you, this change has been made.

ˆP1L22. See specific points about the interpretation in terms of activity cycles.

***See response to Specific Point #2

^P3L10. The authors state that MISR has the potential to distinguish the emission from (a) ash explosions, (b) pulsatory degassing, (c) gas jetting, and (d) explosive activity. May the authors identify those emissions for each MISR data shown in this paper?

***Please see response to Specific Point #2

^P3L12-13. Delete this sentence. See specific points.

***P4L16-23 – Sentences in this section have been edited and rearranged for clarity.

^P3L18-20. This is in general true at the same atmospheric conditions. I suggest to delete this sentence.

***P4L17-26 – Sentences in this section have been rearranged. The referenced sentence forms part of the justification as to why the larger Vulcanian plumes are the primary eruption type that can be observed by MISR.

^P4L17. Explain 'derive proxy particles type'.

***P6L12-14 – This section has been revised for clarity.

^P5L3. Define SSA.

***P7L9 – Thank you, this change has been made. SSA = Single-scattering albedo.

^P5L13. Add the period.

***P7L22 – Thank you, this change has been made.

^P5L15. Add how the size, shape and absorption are retrieved by the MISR RA.

***P6L17-23 – These retrievals are derived through the comparison of radiance recorded in the 9 cameras over 4 spectral bands with a 774-mixture look up table. A brief summary is given, and details of this are included in the references in the second paragraph of Section 2.3.

^P5L20. Add the size retrieved by MISR for large, medium and small particles.

***P7L4-7 – Thank you. The size distributions are given in Table 2, and we have added a note directing readers to this table.

^P8L15-17. Due the very few MISR observation rate (about 4%), this sentence is not well supported.

***P12L7-11 – This section has been revised for clarity. The observation rate is explained by the fact that we can only observe the largest forms of activity displayed by Karymsky; this is now mentioned within the text. We have also added an additional reference to the fact that we would require coincident observations to validate the capability of MISR retrievals.

^P9L14. Add the distance from the volcanic vent.

***P13L20 – Thank you, the distance from the vent has been added to the text.

^P10. Replace 'geological' with 'volcanological' in the sub-chapter 4.2. P10L10. Improve the description of the eruptive phases.

***P14L20 – Thank you, this change has been made to the title. An improved description of eruption phases has been included at the end of the first paragraph of this section (P15L1-8).

^P10L24. I wonder if the shift from effusive to explosive activity is given only by the amount of ash in the atmosphere without taking into account the amount of sulfate/water that could be also high for both activities. May the authors add some references?

***P15L22-P16L6 – Suggested shifts in activity were defined through interpretation of both the MISR plume characteristics and the MODIS thermal output. We plan to provide additional eruptive style constraints in future through the incorporation of UV satellite data, sensitive to SO2 emissions.

^P11L2-5. Clarify this sentence. P11L11. Fig. 4c is lacking.

***P16L14-19 – Thank you, we have clarified this sentence. The figure reference has been updated to the correct figure.

^P11L18. The hypothesis of pyroclastic flows should be justified by published papers or news from web-sites.

***P17L4-9 – Thank you, references have been added to justify: plume dispersion characteristics; thermal anomaly generation following pyroclastic flows; and the low intensity thermal alert report.

^P11L26. Is MISR able to distinguish among sulfate or water vapor plumes? P12L16-17. Clarify.

***P17L13 – The similarity of sulfate and water vapor particles (MISR small class) and radiative properties (spherical, non-absorbing) limit our ability to distinguish these components (see Scollo et al., 2012). This was not something that affected analysis in this case, as the plumes we observed were predominantly ash rich.

^P12L21-22. Clarify.

***P19L1-4 – This sentence introduces the concluding remarks and therefore the rest of the concluding section clarifies the statement.

^P12L25. What is 'volcano's geologic evolution' for?

***P19L7 – This section has been reworded to address the confusion of this statement.

^P12L28. The eruption style cannot be derived from MISR data analysis. See specific points.

***Please see response to specific point #2

^P12L32. Add the classes.

***P19L15-19 – Thank you, this change has been made.

^P13L10-11. I wonder if this analysis is affected by the small percentage of MISR data respect to the eruptions happened in the same period.

***It is likely that information on any smaller/unobserved plumes would improve the interpretation of the internal dynamics of the volcano. Unfortunately, without available data we are unable to posit these variations. Ongoing work with this data will incorporate analyses from multiple volcanoes with varying coverage levels to assess the influence of limited observations. We are also extending the synergistic use of satellite data in volcano monitoring to include SO2 quantities, among others. However, this is outside the scope of the single volcano case presented here.

^P13L14. See specific comments on eruptive cycles.

***Please see response to specific point #2

^P13L18-19. Are MISR data able to discriminate those processes? How? References

***P20L13-18 – This section has been edited to clarify that these processes are being inferred from interpretation of both the MISR plume details and MODIS thermal anomaly record.

^Johnson, J.B., Lees, J.M., Gordeev, E.I. (1998), Degassing Explosions at Karymsky Volcano, Kamchatka, Geophysical Research Letters, DOI: 10.1029/1998GL900102. Ozerov, A., Ispolatov, I., Lees, J. (2003), Modeling Strombolian eruptions of Karymsky volcano, Kamchatka, Russia, Journal of Volcanology and Geothermal Research 122, 265-280. Wen, S., and W. I. Rose (1994), Retrieval of sizes and total masses of particles in volcanic clouds using AVHRR bands 4 and 5, J. Geophys. Res., 99, 5421–5431, doi:10.1029/93JD03340.

***P3-4 – Thank you for suggesting these relevant references; they have been added to the paper.

Figures

ˆFigure 1 is not cited in the text.

***P3L19 – Thank you, this has been corrected.

ˆFigure 2, 3 and 4. Add the scale in km.

***P28-30 – Thank you, the scale has been added.

ˆFigure 7. Add the scale at the bottom of figures. In Fig. 7 B, the plume height of 7 km reached far from the volcanic vent is higher than the height above the vent. Why?

***P33 – Thank you, the scale has been added. A reference to the uplift is included (PL) and is the result of a frontal system moving from the east and causing uplift of the plume and surrounding air mass.

ˆTable 1. Improve the eruption description.

***P24 – The table has been edited to indicate that observations are based on the MISR analysis. Unfortunately, the eruption reports reviewed for these plumes are extremely general and no coincident observations were found in the literature. Therefore, extensive information on each specific eruption was not available for inclusion.

ˆI suggest to move the plots from the supplementary material to the paper and, moreover, add the plots 'height versus distance from the volcanic vent' as retrieved by the MINX software. Finally, the column heights reported in the paper should be above sea level.

***Supplement – The height vs. distance and wind speed plots have been added to the Supplemental Data Unfortunately, the large number and size of each plume retrieval makes it impractical to include all of them in the paper itself. We are grateful that the Supplemental data option allows us to include this much ancillary information for those interested in the details.

Please also note the supplement to this comment:

https://www.atmos-chem-phys-discuss.net/acp-2017-868/acp-2017-868-AC2-supplement.pdf

**Supplement:**

[revised manuscript text omitted]

---

## Author Response (AR2)

Response to editors review comments

*Thank you for your additional comments. Please find below details of the changes that have been made to address these issues. In addition to these specific comments, a review of the paper has been performed with minor copy edits made throughout. We have also altered the title slightly to reduce the redundancy resulting from the previous change suggested by reviewers.*

- In the abstract and elsewhere, please refer to the "2014-2015 Holuhraun eruption". Bárðarbunga is the volcanic system at which subsidence occurred, but the eruption you study is officially referred to as "2014-2015 Holuhraun eruption"
*References to Bárðarbunga in the abstract (L19) and throughout the paper have been changed to Holuhraun.*

- Abstract, line 26: Before reading the entire paper, I found it was not clear me what you mean by 'established sulfate proxy' - I recommend you rewrite this sentence so it's understandable for readers who have not read the entire text and/or are not familiar with Karymsky volcano.
- The last sentence of the abstract is to my mind a little cumbersome and contains little information unless one reads the entire paper. I recommend spelling out what these "similar plume components" are or you state the period prior to 2011 you refer to in terms of similarities.
*P1L25-P2L3 – The last line of the abstract has been completely rewritten for clarity. This includes addressing the changes to the "sulfate-proxy" sentence.*

- Section 5, line 22: strictly speaking Holuhraun was a "sulfur-rich eruption" (although I realise you refer to what you detect)
*P19L19 – I have altered the reference to the overall eruption type of Holuhraun to "sulfur-rich" from "sulfate-rich"*

- Figures:

o Figures 2, 3, 4, and 7 all need to have longitude and latitude coordinates added
*Latitude and longitude lines have been added to Figures 2, 3, 4 and 7.*

o Figure 5: please state the AOD wavelength and add units to each panel where appropriate
*The AOD wavelength has been added to each of the plots in Figure 5. The panels with no units are dimensionless values and therefore have not bee changed.*

o Figure 8: for the Holuhraun inset, I would suggest using the words 'sulfur-rich plumes' to better reflect the entirety of the plume constituents.
*The inset of Figure 8 has been updated as suggested.*

**Karymsky volcano eruptive plume properties based on MISR multi-angle imagery, and volcanological implications**

Verity J. B. Flower[1,2] and Ralph A. Kahn[1]

[revised manuscript text omitted]

Verity 11/28/17 6:29 PM

Flower , Verity J. (GS…, 11/30/17 2:36 PM
Deleted: of the volcano's geologicdevelopment evolution …Despite … for the MISR retrieval climatology, and…l…ed … because the longest MISR wavelength is 866 nm,…from eruption to…can be…,… downwind, and cycles in the activity within the volcanic source.   … [3]

Flower , Verity J. (GS…, 11/30/17 2:38 PM
Deleted: (1 -10 µm size range; 1.28 µm effective radius) type …,…and o… (0.75 µm effective radius)… (0.12 µm effective radius)…The observed -…s…ate…Bárðarbunga…demonstrate that the presence of a sulfate particle proxy in the MISR retrievals (90% MeSpNab, 10% SmSpNab)…s…volatile…along individual plumes provide …s of…indicating   … [4]

Flower , Verity J. (GS…, 11/30/17 2:44 PM

larger particles, with a moderate fraction of the sulfate analog. Particle property variations appear to correlate with changes in the eruption dynamics of individual phases. In 2007, a single eruption phase indicates a shift toward more sulfur-rich, smaller-sized particles as thermal output increased, from which we infer the ascent of more volatile-rich magma, culminating in multiple eruptions, magma depletion, and finally, rapid termination of the eruption cycle. MISR captured only one low-elevation plume at Karymsky between 2007 and 2011, likely resulting from a small dome collapse and pyroclastic flow event.  Plumes generated in 2011 indicate a stabilization of volcanic processes leading to predictable particle evolution throughout the eruption phase. These developments suggest a reduction in the volatile content and alteration of the chemical composition of the feeding magma. The level of absorbing components appears to correlate strongly with normalized eruption day, assessed relative to the earliest thermal anomaly observed during a particular phase; this metric facilitates comparison between eruption phases. Plumes emitted in 2013 and 2015 also appear to follow this pattern of particle development relative to the normalized eruption start date. The consistency of eruption development suggests that the magmatic feeding system at Karymsky stabilized as the eruption waned (2011-2016).

Deductions based upon space-based remote-sensing observations, corroborated by suborbital measurements where possible, illustrate the ability to apply such data to volcanology-from-space studies more broadly. As the satellites provide frequent, planet-wide coverage, there are considerable opportunities for further study, even of remote locations around the globe where *in situ* monitoring is lacking.

**Acknowledgements**

The work of V.J.B. Flower is supported by an appointment to the NASA Postdoctoral Program at the NASA Goddard Space Flight Center, administered by Universities Space Research Association under contract with NASA. The work of R. Kahn is supported in part by NASA's Climate and Radiation Research and Analysis Program under H. Maring, NASA's Atmospheric Composition Program under R. Eckman, and the NASA Earth Observing System's MISR project. We thank James Limbacher for his

assistance in the installation of the MISR 774-mixture Research Aerosol algorithm and his ongoing technical support.

**References**

[revised manuscript text omitted]

Flower , Verity J. (GS…, 2/9/18 11:02 AM